# Towards Efficient Constraint Handling in Neural Solvers for Routing Problems

**Jieyi Bi**[1,5♥]**, Zhiguang Cao**[2]**, Jianan Zhou**[1]**, Wen Song**[3]**, Yaoxin Wu**[4]**, Jie Zhang**[1]**,
Yining Ma**[5†]**, Cathy Wu**[5]

[1]Nanyang Technological University     [2]Singapore Management University
[3]Shandong University     [4]Eindhoven University of Technology     [5]MIT
`jieyi001@e.ntu.edu.sg, zgcao@smu.edu.sg, jianan004@e.ntu.edu.sg,`
`wensong@email.sdu.edu.cn, y.wu2@tue.nl, zhangj@ntu.edu.sg,`
`{yiningma, cathywu}@mit.edu`

## Abstract

Neural solvers have achieved impressive progress in addressing simple routing problems, particularly excelling in computational efficiency. However, their advantages under complex constraints remain nascent, for which current constraint-handling schemes via *feasibility masking* or implicit *feasibility awareness* can be inefficient or inapplicable for hard constraints. In this paper, we present Construct-and-Refine (CaR), the first general and efficient constraint-handling framework for neural routing solvers based on explicit learning-based *feasibility refinement*. Unlike prior construction-search hybrids that target reducing optimality gaps through heavy improvements yet still struggle with hard constraints, CaR achieves efficient constraint handling by designing a joint training framework that guides the construction module to generate diverse and high-quality solutions well-suited for a lightweight improvement process, e.g., 10 steps versus 5k steps in prior work. Moreover, CaR presents the first use of construction-improvement-shared representation, enabling potential knowledge sharing across paradigms by unifying the encoder, especially in more complex constrained scenarios. We evaluate CaR on typical hard routing constraints to showcase its broader applicability. Results demonstrate that CaR achieves superior feasibility, solution quality, and efficiency compared to both classical and neural state-of-the-art solvers. Our code, pre-trained models, and datasets are available at: `https://github.com/jieyibi/CaR-constraint`.

## 1 Introduction

Vehicle Routing Problems (VRPs) often involve complex real-world constraints (Wu et al., 2023). Classic VRP solvers, such as LKH-3 (Helsgaun, 2017) and OR-Tools (Furnon & Perron, 2024), have relied on heuristics carefully designed by human experts to handle these constraints and approximate near-optimal solutions. Recently, Neural Combinatorial Optimization (NCO) methods (Bengio et al., 2021) have offered a different path: they automate solver design with deep learning and exploit GPU-batched inference for high efficiency while ensuring solution quality (Kwon et al., 2020). However, recent work has primarily targeted simple variants, leaving their potential on hard-constrained VRPs underexplored. In those more complex settings, where hand-crafted heuristics often leave certain research gaps, reinforcement learning (RL)-based methods may offer a promising alternative by learning to navigate constraints directly from data. This makes effective constraint handling a key challenge in advancing the broader applicability of NCO to real-world VRPs.

Most RL-based NCO solvers handle constraints via two schemes: *feasibility masking* and implicit *feasibility awareness*. Feasibility masking enforces constraints by excluding invalid actions in the Markov Decision Process (MDP). While effective for simple MDPs, it becomes *intractable* in complex cases where computing mask itself is NP-hard, e.g., with complex local search operators in neural improvement solvers (Ma et al., 2023), or with interdependent constraints in neural construction

---

♥ Work done during the author's internship at MIT.     † Corresponding author.

solvers (Bi et al., 2024), as in Traveling Salesman Problem with Time Windows (TSPTW). Moreover, even when computable, the impact of strict masking in multi-constraint VRPs is often overlooked. As shown in Section 4.1, enforcing strict masks in Capacitated VRP with backhaul, duration, and time-window constraints (CVRPBLTW) can largely hinder RL convergence to a better policy. A recent line of work instead explores *feasibility awareness*, implicitly informing MDP decisions via constraint-related features (Chen et al., 2024), reward shaping (Ma et al., 2023), or approximated learnable masks (Bi et al., 2024). However, they still remain limited: to our knowledge, no single method is generic to be effective across typical hard VRPs (e.g., TSPTW and CVRPBLTW), feasibility and optimality gaps persist, and they often incur substantial inference overheads that undermine the efficiency. This motivates a practical research question: *Can we develop a neural constraint handling framework that is simple, general, and crucially, preserves the efficiency of the NCO solvers?*

To address this, we emphasize learning-based *feasibility refinement* that has been overlooked for neural solvers: rather than purely focusing on enforcing feasibility via masking or implicitly learning feasibility signals by features or reward, we ask whether RL can explicitly refine infeasible solutions in very few post-construction steps while preserving optimality. To realize this, one may consider leveraging existing search techniques, such as random reconstruction (RRC) (Luo et al., 2023), efficient active search (EAS) (Hottung et al., 2022), or hybrid frameworks like collaborative policies (LCP) (Kim et al., 2021), RL4CO (Berto et al., 2025), and NCS (Kong et al., 2024). However, these methods are designed and evaluated to reduce optimality gaps on simple VRPs. When applied to hard VRPs, they can yield 100% infeasibility or rely on a prolonged improvement process that runs for hours and often fails when search steps are reduced during training or inference (see Table 2).

In this paper, we present **Construct-and-Refine (CaR)**, a simple yet effective *feasibility refinement* framework as a first step toward a *general* neural approach for *efficient* constraint handling. CaR introduces an end-to-end joint training framework that unifies a neural construction module with a neural improvement module. By design, it unites the complementary strengths of both paradigms while targeting efficiency: construction provides diverse and high-quality solutions conducive to fast refinement guided by our tailored loss function. Unlike prior hybrids that rely on heavy improvement to reduce optimality gaps, CaR uses fewer refinement steps, which can reduce runtime from hours to minutes or seconds. Moreover, the proposed *feasibility refinement* scheme inherently motivates a novel form of synergized *feasibility awareness*. CaR thus further considers a construction-improvement shared encoder for realizing the cross-paradigm representation learning, enabling potential knowledge sharing, thereby improving performance, particularly for more complex constrained scenarios.

We demonstrate the effectiveness of CaR on VRPs with diverse constraints, with particular emphasis on hard-constrained settings. When feasibility masking is NP-hard (e.g., TSPTW), CaR achieves 6-8× speedups over state-of-the-art neural baselines while improving feasibility and solution quality, even finding feasible solutions that the strong classic solver LKH-3 fails to produce. When masking is tractable but overly restrictive (e.g., multi-constraint CVRPBLTW), CaR shows dominant superiority over both neural and classic solvers. We also provide detailed analysis and ablation studies.

Our contributions are as follows: **1)** We comprehensively analyze the limitations of existing *feasibility masking* and *implicit feasibility awareness* schemes, and introduce the learning-based *feasibility refinement* scheme for efficient constraint handling; **2)** We present Construct-and-Refine (CaR), a simple, general, efficiency-preserving framework that performs *feasibility refinement* via end-to-end joint training that constructs diverse and high-quality solutions well-suited for rapid refinement guided by our tailored losses; **3)** CaR enables novel synergized *feasibility awareness* via cross-paradigm representation learning that further boosts the constraint handling especially in complex cases; **4)** Experiments showcase that CaR can be potentially applicable to enhance most RL-based construction and improvement solvers in solving various hard-constrained VRPs, delivering superior feasibility, solution quality, and efficiency compared to classical and neural state-of-the-art solvers.

## 2   LITERATURE REVIEW

We highlight the difference between CaR and existing neural solvers in Table 1: most existing neural solvers emphasize masking-based feasibility handling, whereas CaR integrates three complementary schemes for broader and more effective constraint handling.

Table 1: Comparison between CaR and other existing neural solvers. CaR is the first to cover both simple and complex VRPs by combining flexible‡ masking, shared-representation to implicitly enhance feasibility awareness, and explicit feasibility refinement.

| Methods | VRP applicability | | Research focus | | Feasibility Masking | Feasibility Awareness | | | Feasibility Refinement |
|---|---|---|---|---|---|---|---|---|---|
| | Simple | Complex | Optimality | Feasibility | | Penalty | Features | Shared Rep. | |
| Most existing neural solvers (e.g. POMO, UDC, LCP, RL4CO, NCS) | ✓ | ✗ | ✓ | ✗ | ✓ | ✗ | ✗ | ✗ | ✗ |
| NeuOpt-GIRE (Ma et al., 2023) | ✓ | ✗ | ✓ | ✗ | ✓ | ✗ | ✓ | ✗ | ✗ |
| Tang et al. (2022) | ✗ | ✓ | ✓ | ✓ | ✓ | ✓ | ✗ | ✗ | ✗ |
| MUSLA (Chen et al., 2024) | ✗ | ✓ | ✓ | ✓ | ✓ | ✗ | ✓ | ✗ | ✗ |
| PIP (Bi et al., 2024) | ✗ | ✓ | ✓ | ✓ | ✓ | ✓ | ✗ | ✗ | ✗ |
| CaR (Ours) | ✓ | ✓ | ✓ | ✓ | ✓‡ | ✓ | ✓ | ✓ | ✓ |

**Neural VRP solvers.** End-to-end neural VRP solvers mainly fall into two paradigms: 1) *Construction solvers* learn to construct solutions from scratch in an autoregressive (AR) fashion. Among them, Attention Model (AM) (Kool et al., 2018) is a milestone for VRPs. POMO (Kwon et al., 2020) further improved AM using diverse rollouts inspired by VRP symmetries. Subsequent studies have advanced AR solvers in inference strategies (Choo et al., 2022), training paradigms (Drakulic et al., 2023; Hottung et al., 2025), scalability (Jin et al., 2023; Hou et al., 2023), robustness (Geisler et al., 2022), and generalization over different distributions (Bi et al., 2022), scales (Zhou et al., 2023; Gao et al., 2024), and constraints (Zhou et al., 2024; Berto et al., 2024); 2) *Improvement solvers* learn to iteratively improve initial solutions, inspired by classic local search such as *k*-opt (Costa et al., 2020; Ma et al., 2023), ruin-and-repair (Hottung & Tierney, 2022; Ma et al., 2022), and crossover (Kim et al., 2023). In general, they achieve near-optimal solutions with prolonged searches. Overall, either construction or improvement solvers mainly focus on simple VRP benchmarks, leaving complex constraint handling underexplored. See Appendix A for further discussion.

**Constraint handling for VRPs.** Existing neural solvers handle constraints mainly through two schemes: *feasibility masking* and implicit *feasibility awareness*. Most methods enforce constraint satisfaction by excluding invalid actions, such as local search moves in neural improvement solvers or node selection in neural construction solvers, via strict feasibility masking. While effective in simple VRPs, masking becomes intractable or ineffective in complex ones (e.g., TSPTW, CVRP-BLTW; see Section 4.1). Recent works have thus explored implicit *feasibility awareness*, enhancing neural policies via reward/penalty-based guidance or feature augmentation, such as Lagrangian reformulation (Tang et al., 2022), reward shaping for infeasible-region exploration (Ma et al., 2023), constraint-related features (Chen et al., 2024), or approximated learnable masks (Bi et al., 2024). However, they often incur high computational cost and still yield high infeasibility. Overall, even with these two constraint handling schemes, no existing solvers is generic enough to be effective across both simple and complex VRPs, motivating a new perspective: explicit *feasibility refinement*.

**Hybrid neural solvers.** Recent works have explored hybridizing construction with policy search or heavy improvement. *One line* extends construction with additional search, e.g., active search (EAS) (Hottung et al., 2022) or beam search (SGBS) (Choo et al., 2022), but these remain confined to the underlying construction policy and struggle with complex constraints. Another line employs reconstruction or decomposition, such as random reconstruction (RRC) (Luo et al., 2023) and collaborative policies (LCP) (Kim et al., 2021). While effective on simple VRPs, these methods often break feasibility when reconstructing or decomposing solutions, simlilar as other divide-and-conquer frameworks (Ye et al., 2024; Zheng et al., 2024), leading to high infeasibility even with penalties. *Another line* combines construction with heavy improvement (e.g., thousands of steps). RL4CO (Berto et al., 2025) couples pretrained POMO (Kwon et al., 2020) and NeuOpt (Ma et al., 2023) only at inference, while NCS (Kong et al., 2024) integrates them via a shared critic. However, both remain limited on hard-constrained VRPs, with NCS yielding 100% infeasibility on TSPTW in our tests. *We differ from them in three key aspects*: 1) we jointly targets feasibility and optimality, rather than only focusing on optimality; 2) we tightly couples construction and refinement via joint training and shared representation learning, enabling knowledge transfer, instead of treating both paradigms separately (Berto et al., 2025) or loosely coupled (Kong et al., 2024); 3) we promote efficient and precise refinement rather than lengthy improvement (e.g., from 5k to 10 steps).

## 3 PRELIMINARIES

**VRP definitions.** We define VRP on a directed graph $\mathcal{G} = (\mathcal{V}, \mathcal{E})$, where $\mathcal{V}$ consists of $n$ customer nodes $\{v_1, \ldots, v_n\}$ and a depot $v_0$ (except TSP variants). Each edge $e(v_i \rightarrow v_j)$ (or $e_{ij}$) $\in \mathcal{E}$ connects

node $v_i$ to $v_j$ ($i \neq j$) with weight given by the 2D Euclidean distance. The objective is to minimize the total cost of a solution subject to variant-specific constraints. Empirically, we distinguish between *simple VRPs* (e.g., CVRP), where feasibility masking is tractable and effective, and *complex VRPs* (e.g., CVRPBLTW with multiple constraints or TSPTW where computing masks is NP-hard since future time-feasibility must be evaluated for all possible actions), where masking is ineffective or intractable. See Appendix B for problem definitions and instance generation details.

**MDP formulations.** We consider neural solvers trained with reinforcement learning (RL), where construction and improvement are formulated as Constrained Markov Decision Process (CMDP), defined by the tuple $(\mathcal{S}, \mathcal{A}, \mathcal{P}, \mathcal{R}, \mathcal{C})$, with $\mathcal{S}$ as the state space, $\mathcal{A}$ as the action space, $\mathcal{P} : \mathcal{S} \times \mathcal{A} \times \mathcal{S} \rightarrow [0, 1]$ as the state transition probability, $\mathcal{R} : \mathcal{S} \times \mathcal{A} \rightarrow \mathbb{R}$ as the reward function, and $\mathcal{C} : \mathcal{S} \times \mathcal{A} \rightarrow \mathbb{R}^m$ as the constraint function penalizing violations of $m$ constraints. The goal is to learn a policy $\pi_\theta : \mathcal{S} \rightarrow \mathcal{P}(\mathcal{A})$ that maximizes the reward while satisfiying the constraint(s), i.e.,

$$\max_\theta \mathcal{J}(\pi_\theta) = \mathbb{E}_{\tau \sim \pi_\theta} [\mathcal{R}(\tau | \mathcal{G})], \text{ s.t. } \pi_\theta \in \Pi_F, \ \Pi_F = \{\pi \in \Pi \, | \, \mathcal{J}_C(\pi_\theta) = \mathbf{0}^m\}, \tag{1}$$

where $\mathcal{J}$ and $\mathcal{J}_C$ are the expected returns of the reward and constraint function for $m$ constraints, respectively; and $\Pi_F$ represents a feasible space for the policy. *Construction* solvers construct solutions sequentially from scratch: state $s_t$ includes the partial solution, vehicle status (e.g., load, time), and unvisited node representations; action $a_t$ selects the next node; and the reward $\mathcal{R}$ is the negative tour cost at completion, $\mathcal{R}(\tau | \mathcal{G}) = -C(\tau)$. *Improvement* solvers learn to refine an existing solution. At step $t$, the state $s_t$ includes the current solution $\tau_t$, the best-so-far solution $\tau_t^*$, and instance features; the action $a_t$ applies an operator (e.g., flexible $k$-opt (Ma et al., 2023), remove-and-reinsert (Ma et al., 2022); see Appendix C). Following the convention (Chen & Tian, 2019), the reward is the cost reduction in the best-so-far solution, i.e., $\mathcal{R}_t = \min[C(\tau_{t-1}^*) - C(\tau_t), 0]$.

**Relaxation of CMDP.** Following Bi et al. (2024), we apply Lagrangian relaxation to train the construction policy by penalizing constraint violations, adding a cost term to the objective in Eq. (1):

$$C(\tau) = \sum_{e_{ij} \in \tau} \left[ C_L(e_{ij}) + \sum_{\eta=1}^{m} C_V^\eta(e_{ij}) \right], \tag{2}$$

where $C_L$ and $C_V$ represent the objective cost (i.e., tour length) and the constraint violations, respectively. For example, if the arrival time $t_j$ to node $v_j$ exceeds the time window ends $u_j$, the node-level cost for time window violation is calculated as $C_V(e_{ij}) = t_j - u_j$. The total violation cost $C_V$ also accounts for the number of nodes with constraint violations to enhance constraint awareness.

## 4 METHODOLOGY

We revisit existing constraint-handling schemes for neural solvers and then introduce our CaR framework, which proposes a new perspective via *feasibility refinement* and further explores shared representations across construction and refinement to enhance *feasibility awareness*.

### 4.1 DISCUSSION OF EXISTING CONSTRAINT HANDLING SCHEMES

*Feasibility masking* excludes invalid actions at each node-selection step in construction MDPs and each local-search step in improvement MDPs. It is widely used in prevailing neural solvers and works well when VRP constraints are simple. For instance, Table 12 shows that removing masking in CVRP increases POMO's optimality gap from 0.86% to 0.92%, confirming its effectiveness. However, masking faces two fundamental challenges in complex VRPs. First, mask computation itself can be *intractable*. For example, in TSPTW evaluating time-interdependent feasibility at each construction step requires checking all future actions, which is NP-hard (Bi et al., 2024); similarly, computing feasible moves for local search operators such as $k$-opt is intractable in improvement solvers (Ma et al., 2023). Second, even when tractable, masks can be overly restrictive in multi-constraint VRPs. In CVRPBLTW, for instance, strict masking filters out more than 60% of nodes (Figure 6), severely limiting the search space and hindering RL convergence toward a high-quality policy (more discussion in Appendix E.1). In these complex VRPs, approximate mask (Bi et al., 2024) or relaxed masks (as seen in POMO* vs. POMO in Table 2) provide partial relief but cannot fully resolve these issues: they may still fail to guarantee feasibility, introduce computational inefficiency, or degrade solution

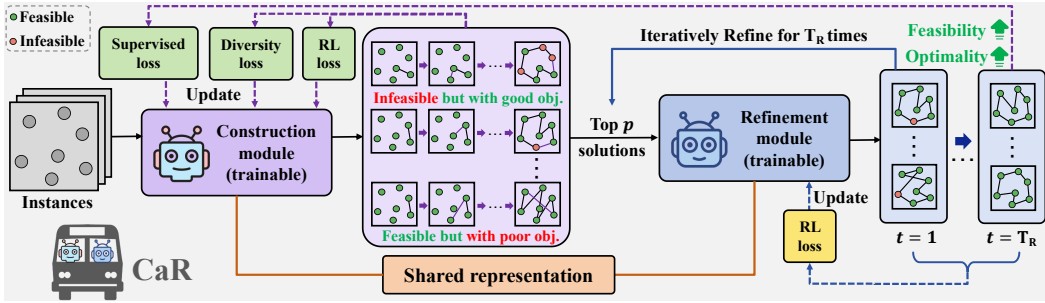

Figure 1: Overall framework of CaR, where the construction module provides diverse and high-quality solutions for the refinement module to generate better solutions.

quality. Beyond masking, recent works have turned to *feasibility awareness*, implicitly guiding MDP decisions via rewards/penalties or constraint-related features. However, the former has been shown in Bi et al. (2024) to lose effectiveness in complex VRPs, while the latter serves only as an auxiliary signal for policy learning. This motivates the need for alternative schemes that explicitly handle feasibility through refinement.

## 4.2 JOINT TRAINING FEASIBILITY REFINEMENT WITH A CONSTRUCTION MODULE

To enable efficient feasibility refinement, we leverage the efficiency of construction and propose our CaR framework. CaR jointly trains construction and refinement by guiding construction module to generate diverse, high-quality solutions that are well-suited for rapid refinement with tailored losses. To integrate construction and refinement effectively, we design a joint training framework that optimizes both processes simultaneously in each gradient step, allowing them to co-evolve. As illustrated in Figure 1, for each batch of training instances $\mathcal{G}$, the construction module first generates a small set of diverse, high-quality initial solutions in parallel. These solutions are then refined by a lightweight neural improvement process within $T_R$ steps (i.e., $T_R = 10$ vs. 5k in classic improvement methods), enabling rapid enhancement of high-potential candidates. The refined outputs then supervise construction, promoting collaborative correction of infeasibility and sub-optimality.

**Construction policy loss.** The policy $\pi_\theta^C$ is trained by REINFORCE (Williams, 1992) with loss:

$$\mathcal{L}_{\text{RL}}^C = \frac{1}{S} \sum_{i=1}^{S} \left[ \left( \mathcal{R}(\tau_i) - \frac{1}{S} \sum_{j=1}^{S} \mathcal{R}(\tau_j) \right) \log \pi_\theta^C(\tau_i) \right], \tag{3}$$

where the solution probability is factorized as $\pi_\theta^C(\tau) = \prod_{t=1}^{|\tau|} \pi_\theta^C(e_t \mid \tau_{<t})$, with $\tau_{<t}$ denoting the partial solution prior to selecting edge $e_t$ at step $t$. We employ a group baseline with diverse rollouts to reduce the variance. For simpler variants like CVRP, $S$ solutions are generated via POMO's multi-start strategy (Kwon et al., 2020), while for time-constrained variants (e.g., TSPTW and CVRPBLTW), we sample $S$ solutions to avoid infeasibility (Hottung et al., 2025; Bi et al., 2024).

**Tailored losses in construction module.** To compensate for reduced diversity due to the removal of the multi-start mechanism and to enhance the diversity of initial constructed solutions for refinement, we introduce an auxiliary entropy-based diversity loss:

$$\mathcal{L}_{\text{DIV}} = - \sum_{t=1}^{|\tau|} \pi_\theta^C(e_t \mid \tau_{<t}) \log \pi_\theta^C(e_t \mid \tau_{<t}), \tag{4}$$

which largely encourages policy exploration during RL training. To avoid inefficiency, we evaluate candidates using the cost in Eq. (2), and only feed the top $p$ high-quality candidates to subsequent refinement. If the refinement module improves a constructed solution (indicated by $\mathbb{I} = 1$), the best-refined solution $\tau^*$ is used as a pseudo ground truth to supervise $\pi_\theta^C$:

$$\mathcal{L}_{\text{SL}} = -\mathbb{I} \cdot \sum_{t=1}^{|\tau^*|} \log \pi_\theta^C(e_t^* \mid \tau_{<t}^*), \tag{5}$$

where $\mathbb{I}$ indicates whether such refinement led to improvement of the feasibility and objective. The final construction loss integrates three components, i.e., $\mathcal{L}(\theta^C) = \mathcal{L}_{\text{RL}}^C + \alpha_1 \mathcal{L}_{\text{DIV}} + \alpha_2 \mathcal{L}_{\text{SL}}$.

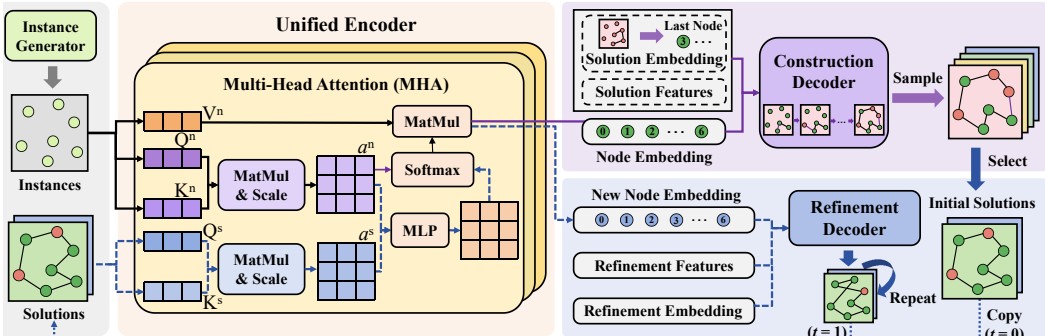

Figure 2: Overview of the unified network architecture in CaR. Blue dashed arrows indicate information flow specific to refinement, while purple dashed arrows indicate flow exclusive to construction.

**Relaxation of short-horizon CMDP for efficient refinement.** Building on the success of relaxed CMDP formulations for construction (Bi et al., 2024), we extend it to the refinement process as well. Unlike prior work (Kong et al., 2024; Ma et al., 2023), which models improvement as an infinite-horizon MDP under the assumption that prolonged runtime is acceptable (more discussion of short-horizon MDP design in Appendix E.5), we adopt a short-horizon rollout limit $T_R$, treating each step equally, consistent with CaR's efficient refinement design.

**Refinement policy loss.** The refinement policy $\pi_\theta^R$ iteratively improves solutions over $T_R$ steps, with probability of the refined solution at step $t$ is factorized as $\pi_\theta^R(\tau_t) = \prod_{\kappa=1}^{K} \pi_\theta^R(a_\kappa | a_{<\kappa}, \tau_{t-1})$, where $K$ denotes the total number of sequential refinement moves/actions, with further details in Appendix C. The RL loss $\mathcal{L}_{RL}^R(t)$ for refinement is computed at each step $t$ using the REINFORCE algorithm in Eq. (3), where $S$ is replaced by $p$, since only $p$ solutions are refined. The final refinement loss is defined as the average across all $T_R$ steps: $\mathcal{L}(\theta^R) = \frac{1}{T_R} \sum_{t=1}^{T_R} \mathcal{L}_{RL}^R(t)$, encouraging each refinement step to contribute meaningfully and improving overall refinement efficiency. The joint training loss combines the above two losses, i.e., $\mathcal{L}(\theta) = \mathcal{L}(\theta^C) + \omega \mathcal{L}(\theta^R)$, where $\omega$ balances their scales. Such joint loss promotes information exchange between modules, enhancing synergy in collaboratively handling complex constraints.

## 4.3 CROSS-PARADIGM REPRESENTATION LEARNING FOR FEASIBILITY AWARENESS

Beyond implicit *feasibility awareness* via features or rewards/penalties, our *feasibility refinement* naturally strengthens awareness through cross-paradigm representation learning. To further reduce overhead and promote synergy, we explore shared encoders for knowledge transfer between construction and refinement, especially in hard-constrained VRPs.

**Shared representation across paradigms via a unified encoder.** Given an instance batch $\{\mathcal{G}_i\}_{i=1}^{B}$, both paradigms learns to obtain high-dimensional node embeddings $h_i$ via encoders. For CVRPBLTW, each node $v_i$ is represented by its coordinates, demand (i.e., linehaul or backhaul), time window, and duration limit, i.e., $f_i^n = \{x_i, y_i, q_i, l_i, u_i, \ell\}$. Unlike construction, refinement also incorporates solution features by encoding the sequential structure via positional information. To support both paradigms, we use a shared 6-layer Transformer encoder (Kwon et al., 2020) with multi-head attention. Positional encoding, required only in refinement, is injected using cyclic positional encoding via the synthesis attention mechanism (Ma et al., 2022). As shown in Figure 2, a multi-layer perceptron (MLP) fuses node-level attention scores $a^n$ and solution-level scores $a^s$ from positional embedding vectors via element-wise aggregation.

**Decoder.** The decoder generates action probabilities from node representations, selecting the next node for construction or the modification for refinement. In CaR, we retain the original neural solver designs when applied to *one* construction and *one* improvement at a time. To validate generality, we experiment with two construction backbones, POMO (Kwon et al., 2020) and PIP (Bi et al., 2024), and two refinement backbones, NeuOpt (Ma et al., 2023) and N2S (Ma et al., 2022). To adapt improvement solvers for new variants (e.g., TSPTW, CVRPBLTW), we follow their original design and introduce variant-specific features, such as refinement history and node-level feasibility information (see Appendix C for details), to enhance constraint awareness. While we also explored

Table 2: Results on complex VRPs: best are **bolded**; best within 1 min are shaded to show solver efficiency.

| | Method | #Params | Paradigm § | n=50 | | | | n=100 | | | |
|---|---|---|---|---|---|---|---|---|---|---|---|
| | | | | Obj. ↓ | Gap ↓ | Infsb% ↓ | Time | Obj. ↓ | Gap ↓ | Infsb% ↓ | Time |
| **TSPTW** | LKH-3 (max trials = 100) | / | I | 25.590 | 0.004% | 11.88% | 7m | 46.625 | 0.103% | 31.05% | 27m |
| | LKH-3 (max trials = 10000) | / | I | 25.611 | ◇ | 0.12% | 7h | 46.858 | ◇ | 0.13% | 1.4d |
| | OR-Tools† | / | I | 25.763 | -0.001% | 65.72% | 2.4h | 46.424 | 0.026% | 97.45% | 12m |
| | Greedy-L | / | C | / | / | 100.00% | 21.8s | / | / | 100.00% | 1m |
| | Greedy-C | / | C | 26.394 | 1.534% | 72.55% | 4.5s | 51.945 | 9.651% | 99.85% | 11.4s |
| | POMO | 1.25M | L2C-S | / | / | 100.00% | 4s | / | / | 100.00% | 14s |
| | POMO* | 1.25M | L2C-S | 26.222 | 1.635% | 37.27% | 4s | 47.249 | 1.959% | 38.22% | 14s |
| | POMO* + PIP (greedy) | 1.25M | L2C-S | 25.657 | 0.177% | 2.67% | 7s | 47.372 | 1.223% | 6.96% | 32s |
| | POMO* + PIP (sample 10) | 1.25M | L2C-S | 25.650 | 0.152% | 1.87% | 1m | 47.291 | 1.026% | 4.47% | 4.7m |
| | UDC* (RRC = 250) | 1.56M | L2C-S | / | / | 100.00% | 2.4h | / | / | 100.00% | 4.9h |
| | NeuOpt-GIRE *‡ ($T$ = 1k) | 0.69M | L2I-S | 25.627 | 0.061% | 0.19% | 2.3m | 47.011 | 0.336% | 0.13% | 5.9m |
| | NeuOpt-GIRE *‡ ($T$ = 2k) | 0.69M | L2I-S | 25.621 | 0.044% | 0.04% | 4.6m | 46.955 | 0.215% | 0.06% | 11.8m |
| | NeuOpt-GIRE *‡ ($T$ = 5k) | 0.69M | L2I-S | 25.617 | 0.028% | 0.02% | 11.6m | 46.913 | **0.123%** | 0.02% | 30m |
| | NCS *‡ | 1.64M | L2(C+I)-S | / | / | 100.00% | 11.6m | / | / | 100.00% | 30m |
| | **CaR-POMO ($T_R$ = 5) (Ours)** | 1.64M | L2(C+I)-S | 25.619 | 0.034% | 0.02% | 15s | 47.278 | 1.065% | 4.20% | 36s |
| | **CaR-POMO ($T_R$ = 10) (Ours)** | 1.64M | L2(C+I)-S | 25.615 | 0.020% | 0.01% | 27s | 47.074 | 0.581% | 2.77% | 1.1m |
| | **CaR-POMO ($T_R$ = 20) (Ours)** | 1.64M | L2(C+I)-S | 25.614 | 0.014% | 0.01% | 51s | 47.001 | 0.406% | 2.34% | 2.1m |
| | **CaR-PIP ($T_R$ = 5) (Ours)** | 1.64M | L2(C+I)-S | 25.613 | 0.010% | 0.02% | 17s | 47.000 | 0.315% | 0.10% | 58s |
| | **CaR-PIP ($T_R$ = 10) (Ours)** | 1.64M | L2(C+I)-S | 25.612 | 0.006% | 0.01% | 29s | 46.945 | 0.191% | 0.03% | 1.4m |
| | **CaR-PIP ($T_R$ = 20) (Ours)** | 1.64M | L2(C+I)-S | 25.612 | **0.005%** | **0.00%** | 52s | 46.923 | 0.146% | 0.02% | 2.4m |
| **CVRPBLTW** | OR-Tools (short) | / | I | 14.890 | 1.402% | 0.00% | 10.4m | 25.979 | 2.518% | 0.00% | 20.8m |
| | OR-Tools (long) | / | I | 14.677 | ◇ | 0.00% | 1.7h | 25.342 | ◇ | 0.00% | 3.5h |
| | POMO | 1.25M | L2C-S | 15.999 | 9.169% | 0.00% | 2s | 27.046 | 7.004% | 0.00% | 4s |
| | POMO-MTL | 1.25M | L2C-M | 15.980 | 9.035% | 0.00% | 2s | 27.247 | 7.746% | 0.00% | 7s |
| | MVMoE | 3.68M | L2C-M | 15.945 | 8.775% | 0.00% | 3s | 27.142 | 7.332% | 0.00% | 10s |
| | ReLD-MoEL | 3.68M | L2C-M | 15.925 | 8.623% | 0.00% | 3s | 27.044 | 6.915% | 0.00% | 9s |
| | POMO+EAS+SGBS* (short) | 1.25M | L2C-S | 15.386 | 4.831% | 0.00% | 25s | 26.005 | 2.616% | 0.00% | 2.3m |
| | POMO+EAS+SGBS* (long) | 1.25M | L2C-S | 15.156 | 3.263% | 0.00% | 10.3m | 25.558 | 0.854% | 0.00% | 1h |
| | NeuOpt-GIRE *‡ ($T$ = 1k) | 0.69M | L2I-S | 14.521 | 1.329% | 33.80% | 1.1m | 24.597 | 3.390% | 51.20% | 2.5m |
| | NeuOpt-GIRE *‡ ($T$ = 2k) | 0.69M | L2I-S | 14.352 | -0.031% | 31.00% | 2.2m | 24.365 | 0.875% | 44.50% | 5.1m |
| | NeuOpt-GIRE *‡ ($T$ = 5k) | 0.69M | L2I-S | 14.201 | **-1.163%** | 27.30% | 5.5m | 24.038 | -1.541% | 39.10% | 12.7m |
| | **POMO* (Ours)** | 1.25M | L2C-S | 14.873 | 2.310% | 0.00% | 2s | 24.592 | -1.645% | 0.00% | 4s |
| | **CaR ($k$-opt) ($T_R$ = 5) (Ours)** | 1.64M | L2(C+I)-S | 14.872 | 2.271% | 0.00% | 3s | 24.597 | -1.674% | 0.00% | 5s |
| | **CaR ($k$-opt) ($T_R$ = 10) (Ours)** | 1.64M | L2(C+I)-S | 14.865 | 2.227% | 0.00% | 4s | 24.589 | -1.707% | 0.00% | 9s |
| | **CaR ($k$-opt) ($T_R$ = 20) (Ours)** | 1.64M | L2(C+I)-S | 14.844 | 2.114% | 0.00% | 8s | 24.585 | -1.724% | 0.00% | 17s |
| | **CaR (R&R) ($T_R$ = 5) (Ours)** | 1.72M | L2(C+I)-S | 14.725 | 1.328% | 0.00% | 3s | 24.552 | -1.835% | 0.00% | 6s |
| | **CaR (R&R) ($T_R$ = 10) (Ours)** | 1.72M | L2(C+I)-S | 14.661 | 0.878% | 0.00% | 5s | 24.474 | -2.149% | 0.00% | 10s |
| | **CaR (R&R) ($T_R$ = 20) (Ours)** | 1.72M | L2(C+I)-S | 14.601 | 0.463% | **0.00%** | 10s | 24.400 | **-2.448%** | **0.00%** | 19s |

§ The abbreviations refer to: **I** – Improvement; **L2C** – Learning to Construct; **L2I** – Learning to Improve; **S** – Single-task solver; **M** – Multi-task solver.
† OR-Tools presolves before search; if it detects infeasibility, it terminates immediately, making runtime shorter than preset.

a unified decoder, results in Figure 8 show degraded performance, suggesting that while a shared encoder benefits representation learning, separate decoders remain important for paradigm-specific optimization – an insight for future reference.

## 5 EXPERIMENTS

We now evaluate our proposed Construct-and-Refine (CaR) framework in handling hard-constrained TSPTW and CVRPBLTW instances. We also provide detailed analysis and ablation studies.

**Experimental settings.** Training instances are generated on the fly as in (Zhou et al., 2024; Bi et al., 2024) (see Appendix B). For TSPTW, we mainly focus on the hard variants in (Bi et al., 2024). All experiments are conducted on problem sizes $n$ = 50 and 100, following established benchmarks (Kool et al., 2018; Wu et al., 2021). Models are trained with 20,000 instances per epoch for 5,000 epochs with a batch size of 128 (Zhou et al., 2024). We set $T_R$ = 5 during training. During inference, 8× augmentation (Kwon et al., 2020) is used to construct initial solutions, followed by $T_R$-step refinement. Hyper-parameters are detailed in Appendix D.1. All experiments are conducted on servers equipped with NVIDIA GeForce RTX 4090 GPUs and Intel(R) Core i9-10940X CPUs at 3.30GHz. Our code, pre-trained models, and datasets will be publicly released upon acceptance.

**Baseline.** We compare our CaR framework with state-of-the-art (*SoTA*) classic and neural VRP solvers. *Classic solvers* include 1) LKH-3 (Helsgaun, 2017); 2) OR-Tools (Furnon & Perron, 2024); 3) Greedy heuristics, minimizing stepwise tour length (L) and constraint violation (C); and 4) HGS (Vidal et al., 2012). *Neural solvers* include 1) construction solvers such as the single-task solvers AM (Kool et al., 2018), POMO (Kwon et al., 2020) (+EAS (Hottung et al., 2022) + SGBS (Choo et al., 2022)), BQ-NCO (Drakulic et al., 2023), LEHD (+RRC) (Luo et al., 2023), UDC (+RRC) (Zheng et al., 2024), InViT (Fang et al., 2024), PIP (Bi et al., 2024) and PolyNet (Hottung et al., 2025), as well as the multi-task solvers POMO-MTL (Liu et al., 2024), MVMoE (Zhou et al., 2024)

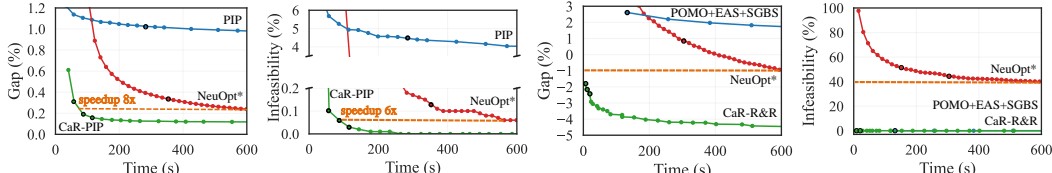

Figure 3: Performance over time on TSPTW-100 (left) and CVRPBLTW-100 (right). For CVRP-BLTW, POMO+EAS+SGBS and CaR always achieve 0% infeasibility due to feasibility guarantee by masking. Dots with black circles represent results reported in Table 2.

and ReLD-MoEL (Huang et al., 2025); 2) the improvement solver NeuOpt-GIRE (Ma et al., 2023); and 3) hybrid solvers, including LCP (Kim et al., 2021). All construction and improvement methods are trained for ~780k (Zhou et al., 2024) (except UDC) and ~600k gradient steps (Ma et al., 2023), respectively, with all methods observed to be converged. For fair comparisons, we use pre-trained models. We note that many baselines were originally designed for simple VRPs (e.g., CVRP) and are not directly applicable to complex variants such as TSPTW and CVRPBLTW. To ensure fair comparison, we upgrade the SoTA neural solvers NeuOpt-GIRE (Ma et al., 2023), UDC (Zheng et al., 2024), and POMO+EAS+SGBS (Choo et al., 2022) using our design enhancements: the relaxed CMDP formulation in Eq. (2) (marked with *) and the proposed solution-level features (marked with ‡; Appendix C). Baseline selection and implementation details are provided in Appendix D.2.

**Evaluation metrics.** We evaluate performance using the metrics below: 1) average solution length (Obj.), representing the mean length of the best feasible solutions; 2) average optimality gap (Gap), measuring the difference between the best feasible solutions and (near-)optimal solutions obtained by the best-performing classic solvers (LKH-3 for TSPTW, OR-Tools for CVRPBLTW, and HGS for CVRP, marked with ⋄); 3) total inference time (Time) taken to solve 10,000 instances for TSPTW or 1,000 instances for CVRPBLTW and CVRP, parallelized on a single GPU; and 4) average infeasible solution ratio (Infsb%) on the best solutions after construction and refinement per instance.

## 5.1 MODEL PERFORMANCE ON VRPS WITH VARYING CONSTRAINT COMPLEXITIES

**TSPTW results.** We first test CaR on TSPTW, where most neural solvers fail due to their reliance but lack of feasibility masking. We use two construction backbones: the lightweight POMO* and the heavier but more effective PIP, with CaR-POMO training 1.42× faster at $n = 50$ and 1.65× faster at $n = 100$ than CaR-PIP. As shown in Table 2, CaR-POMO consistently outperforms PIP in both solution quality and feasibility. On TSPTW-50, CaR reduces infeasibility to 0.00%, outperforming the best construction baseline PIP (1.87%) and our upgraded *SoTA* improvement solver NeuOpt-GIRE* (0.02%). In terms of optimality, CaR nearly matches LKH-3, achieving a minimal gap of 0.005%. On TSPTW-100, CaR lowers PIP's 4.67% infeasibility to 0.02% and reduces the gap from 1.030% to 0.146%. While NeuOpt* improves with extended search (up to 30 minutes), CaR achieves competitive results with an 8× speedup within a runtime budget of 10 minutes (Figure 3), which aligns with CaR's aim of efficiency. Notably, CaR surpasses LKH-3 and finds feasible solutions even when it fails, highlighting CaR's strength under complex constraints. Moreover, we present the case study of CaR's refinement trajectories in Appendix E.6 to show how CaR intelligently refines solutions for feasibility and optimality.

**CVRPBLTW results.** On complex CVRPBLTW, feasibility masking filters out over 60% of nodes, severely limiting the search space (Figure 6). Interestingly, removing these masks and applying the relaxed CMDP in POMO significantly improves performance (e.g., CVRPBLTW-100: from 7.004% to -1.645% in Table 2). Unlike TSPTW, NeuOpt* fails in CVRPBLTW with 27-51% infeasibility, while CaR guarantees feasibility as other construction solvers. We compare CaR with the best single-paradigm solvers in Figure 3. CaR achieves best area under the curve, indicating superior efficiency and effectiveness. We also validate CaR with $k$-opt and R&R, where R&R performs better (-2.448% vs. -1.724%) due to a finer-grained search better suited

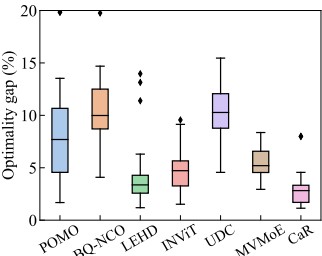

Figure 4: CVRPLIB results.

Table 3: Results on TSPDL-50.

| Method | Gap↓ | Infsb%↓ | Time |
|---|---|---|---|
| PIP (greedy) | 3.122% | 2.12% | 9s |
| PIP (sample) | 2.630% | 1.86% | 60s |
| CaR-PIP | **2.190%** | **0.26%** | 53s |

Table 4: Results on SOP-50.

| Method | SOP Variant 1 | | SOP Variant 2 | |
|---|---|---|---|---|
| | Obj. ↓ | Gap ↓ | Obj. ↓ | Gap ↓ |
| LKH-3 | 14.732 | ◇ | 16.302 | ◇ |
| POMO | 14.943 | 1.436% | 16.376 | 0.463% |
| CaR | 14.831 | **0.676%** | 16.316 | **0.084%** |

Table 5: Effects of joint training on TSPTW-50 under fixed budget. Construction cannot be trained with LKH-3 or pretrained improvement without extra design. See Appendix E.5 for more.

| Joint train | Construction | Improvement | Gap↓ | Infsb%↓ |
|---|---|---|---|---|
| × | Random | LKH-3 | 0.011% | 60.66% |
| × | Random | Pretrained NeuOpt* (long) | / | 100.00% |
| × | Random | Trainable NeuOpt* (short) | / | 100.00% |
| × | Pretrained PIP | LKH-3 | **0.003%** | 0.20% |
| × | Pretrained PIP | Pretrained NeuOpt* (long) | 0.134% | 0.79% |
| × | Pretrained PIP | Trainable NeuOpt* (short) | 0.172% | 2.59% |
| ✓ (CaR) | Trainable PIP | Trainable NeuOpt* (short) | 0.005% | **0.00%** |

to multi-constraints variants. Lastly, we observe that the performance of multi-task neural solvers is largely bottlenecked by their backbone (e.g., POMO), suggesting CaR can serve as a stronger backbone for future NCO foundation models.

**Results on other VRP variants.** We also evaluate CaR on TSP with draft limit (TSPDL), sequential ordering problem (SOP) with discrete precedence constraints and CVRP. Results in Table 3, Table 4 and Appendix E.2 show that CaR consistently delivers competitive results while maintaining efficiency of the neural solvers across different constrained VRPs. To test CaR's generalization across scales, distributions, and simpler VRP variants, we apply it to CVRP and evaluate on CVRPLIB. As shown in Figure 4, CaR achieves strong performance and generalization. Notably, it is the first neural solver to efficiently handle constraints of varying complexity, unlike prior solvers that are either variant-specific (e.g., Bi et al. (2024)) or inefficient (e.g., Ma et al. (2023)).

## 5.2 Effects of the joint training framework

We assess combinations of different construction (random or pretrained PIP) and improvement methods (LKH-3, pretrained long-horizon NeuOpt, and our short-horizon NeuOpt). Table 5 shows that naive combinations yield only marginal gains over PIP alone (0.172% vs. 0.177% gap; 2.59% vs. 2.67% infeasibility). Thus, simply substituting random initialization with a high-quality one is insufficient. However, CaR's joint training unlocks significant performance gains. Despite similar construction quality between pretrained PIP and CaR-PIP (results not shown), the latter exhibits markedly superior refinement. This confirms a non-trivial synergy in CaR, where initialization is specifically tailored for the subsequent refinement.

## 5.3 Effects of the diversification signal $\mathcal{L}_{DIV}$

We investigate the impact of the diversity loss (Eq. 4) in our joint training framework. Recall that while multi-start mechanisms improve performance on simpler problems like CVRP (as shown in Table 12), they often degrade performance on complex constraints like time windows (Bi et al., 2024). Hence, we sample $S$ start nodes to calculate the group baseline in the construction module rather than enumerating all nodes. To compensate for the resulting loss of diversity, we introduce

Table 6: Effects of $\mathcal{L}_{\text{DIV}}$.

| Problem | $\mathcal{L}_{\text{DIV}}$ | Gap ↓ | Infsb% ↓ |
|---|---|---|---|
| **CVRP** | × | 0.325% | **0.00%** |
| | ✓ | **0.259%** | **0.00%** |
| **TSPTW** | × | 0.421% | 0.58% |
| | ✓ | **0.014%** | **0.01%** |

$\mathcal{L}_{\text{DIV}}$. Results in Table 6 show that adding $\mathcal{L}_{\text{DIV}}$ improves overall performance, even when multi-start is enabled in simple CVRP. To verify the diversity increment, we quantify the solution diversity between the constructed solutions of CaR (trained with $\mathcal{L}_{\text{DIV}}$) and PIP (without $\mathcal{L}_{\text{DIV}}$). As shown in Appendix E.3, adding $\mathcal{L}_{\text{DIV}}$ increases the diversity of CaR's constructed solutions by ∼15% over the backbone, confirming that $\mathcal{L}_{\text{DIV}}$ effectively increases solution diversity and enhances performance.

## 5.4 Effects of the supervised signal $\mathcal{L}_{SL}$

We evaluate the impact of the supervised loss $\mathcal{L}_{\text{SL}}$ (Eq. 5) in CaR using two construction backbones: POMO and PIP. Results in Table 7 indicate that the efficacy of the supervised loss depends on the strength of the underlying backbone. For the strong backbone (PIP), which is already highly optimized, with the gap of 0.177% and infeasibility rate of 2.67% (as in Table 2), adding $\mathcal{L}_{\text{SL}}$

Table 7: Effects of $\mathcal{L}_{\text{SL}}$.

| Backbones | $\mathcal{L}_{\text{SL}}$ | Gap ↓ | Infsb% ↓ |
|---|---|---|---|
| CaR-POMO | × | 0.136% | 0.29% |
| CaR-POMO | ✓ | **0.014%** | **0.01%** |
| CaR-PIP | × | 0.006% | 0.01% |
| CaR-PIP | ✓ | **0.005%** | **0.00%** |

Table 8: Effects of shared representation on TSPTW.

| Shared Rep. | #Params | TSPTW-50 Hard | | TSPTW-100 Medium | |
|---|---|---|---|---|---|
| | | Gap ↓ | Infsb% ↓ | Gap ↓ | Infsb% ↓ |
| × | 2.8M | 0.199% | 0.68% | 7.589% | 0.00% |
| ✓ | 1.6M | **0.014%** | **0.01%** | **5.815%** | **0.00%** |

Table 9: Generalization results on CVRPBLTW-200. The runtime for OR-Tools is 1.8h.

| Method | Train Scale | Test Scale | Gap ↓ | Infsb% ↓ | Time |
|---|---|---|---|---|---|
| POMO*+EAS+SGBS (Choo et al., 2022) | 100 | 200 | 4.51% | **0.00%** | 2.1m |
| NeuOpt* (Ma et al., 2023) | 100 | 200 | 6.37% | 75.78% | 17m |
| CaR (R&R, $T_R = 20$) | 100 | 200 | **2.09%** | **0.00%** | **10s** |

Table 10: Generalization results on TSPTW-100 Hard with different time windows tightness.

| Method | Train | Test | Gap ↓ | Infsb% ↓ | Time |
|---|---|---|---|---|---|
| PIP (sample 5) | tight | tight | 0.26% | 4.62% | 2.3m |
| PIP (sample 5) | loose | tight | 0.03% | 31.37% | 2.3m |
| CaR-PIP ($T_R = 20$) | loose | tight | **0.02%** | **3.52%** | 2.3m |

yields only marginal gains; specifically, the gap improves slightly from 0.006% to 0.005%, and the infeasibility rate improves from 0.01% to 0.00%. In contrast, for the weaker backbone (POMO*), where the standalone model performs poorly, with the optimality gap of 1.959% and infeasibility rate of 38.22%, the supervised signal is critical. In this case, joint training with $\mathcal{L}_{\text{SL}}$ markedly improves performance, reducing the gap from 0.136% to 0.014%, and the infeasibility from 0.29% to 0.01%.

## 5.5 EFFECTS OF THE SHARED REPRESENTATION

We study the effect of our shared representation via a unified encoder. As shown in Table 8, CaR with shared representation performs better on both hard-constrained cases (TSPTW-50 Hard and TSPTW-100 Medium), indicating improved knowledge transfer across paradigms. Compared with using separate encoders and decoders, which share only the constructed solutions, CaR also reduces the number of learnable parameters. See Appendix E.4 for further analysis.

## 5.6 GENERALIZATION RESULTS

We now evaluate CaR's generalization performance across problem scales and constraint hardness. Regarding cross-scale generalization, where CaR is trained on CVRPBLTW-100 and tested on CVRPBLTW-200, results show that CaR significantly outperforms other SoTA neural baselines, maintaining 0% infeasibility and the lowest optimality gap within 10 seconds. Furthermore, for cross-constraint generalization, Table 10 demonstrates that CaR trained on loose constraints even outperforms PIP trained on tight constraints (see Appendix B.1 for data generation details).

## 6 CONCLUSION

This paper proposes Construct-and-Refine (CaR), the first neural framework to handle constraints through a new explicit feasibility refinement scheme, extending beyond feasibility masking and implicit feasibility awareness. CaR jointly learns to construct diverse, high-quality solutions and refine them with a lightweight improvement module, enabling *efficient* constraint satisfaction. We also explore shared encoders for cross-paradigm representation learning. To best of our knowledge, CaR is the first neural solvers applicable and effective on VRPs with varying constraint hardness. Future work includes 1) integrating CaR with diverse backbone solvers, 2) applying to more constrained VRPs or even broader COPs, e.g. scheduling (Zhang et al., 2020; 2024; Andelfinger et al., 2025), 3) improving scalability, 4) studying its theoretical properties, e.g. learnability (Yuan et al., 2022), convergence (Thoma et al., 2024), or generalization bound (Duan et al., 2021), and 5) developing foundation NCO models based on the insights of cross-paradigm representation in this paper.

## ACKNOWLEDGMENTS

This research is supported by the National Research Foundation, Singapore under its AI Singapore AI Research Fundamental Research Collaborative (US-NSF Researcher Call) (AISG Award No: AISG3-RP-2025-036-USNSF) and its AI Singapore Programme (AISG Award No: AISG3-RP-2022-031). Any opinions, findings and conclusions or recommendations expressed in this material are those of the author(s) and do not reflect the views of National Research Foundation, Singapore. We would like to thank the anonymous reviewers and (S)ACs of ICLR 2026 for their constructive comments and service to the community.

## REPRODUCIBILITY STATEMENT

We have made every effort to ensure the reproducibility of our results. The datasets used in our experiments are publicly available or generated following standard protocols, with all instance generation details provided in Appendix B. The model architecture is described in Appendix C, with hyperparameters and training/inference configurations detailed in Appendix D.1. Implementation details and the selection criteria for baselines are provided in Appendix D.2. Additional ablation studies and robustness checks are reported in Section 5 and Appendix E. Our code, pre-trained models, and datasets are available at: https://github.com/jieyibi/CaR-constraint.

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

# Appendix

## A    Detailed literature review: neural VRP solvers

Generally, neural VRP solvers can be broadly categorized into two paradigms: construction solvers and improvement solvers.

1) *Construction-based solvers* learn to construct solutions from scratch in an end-to-end fashion. Vinyals et al. (2015) introduced the Pointer Network (PtrNet), leveraging Recurrent Neural Networks (RNN) to solve the Traveling Salesman Problem (TSP) in a supervised manner. Building on this, Bello et al. (2017) explored reinforcement learning (RL) training for PtrNet. Nazari et al. (2018) extended the approach to solve CVRP in an autoregressive (AR) way. Among AR solvers, the Attention Model (AM) (Kool et al., 2018) stands out as a milestone to solve multiple VRPs. This was further advanced by the Policy Optimization with Multiple Optima (POMO) (Kwon et al., 2020), which leverages diverse rollouts inspired by the symmetry properties of VRP solutions. Subsequently, numerous studies have advanced AR solvers in various perspectives, such as inference strategies (Hottung et al., 2022; Choo et al., 2022; Sun et al., 2023), training paradigms (Drakulic et al., 2023; Chalumeau et al., 2023; Grinsztajn et al., 2023; Luo et al., 2023; 2025), interpretability (Kikuta et al., 2024), scalability (Zong et al., 2022; Jin et al., 2023; Hou et al., 2023; Fitzpatrick et al., 2024), robustness (Geisler et al., 2022; Xiao et al., 2024), benchmarking (Thyssens et al., 2023), and generalization over different distributions (Zhang et al., 2022; Bi et al., 2022; Jiang et al., 2022), scales (Zhou et al., 2023; Gao et al., 2024; Fang et al., 2024), and constraints (Lu et al., 2023; Wang & Yu, 2023; Zhou et al., 2024; Liu et al., 2024; Berto et al., 2024; Lin et al., 2024). Beyond AR solvers, another line of research predicts heatmaps in a non-autoregressive (NAR) manner to represent edge probabilities for optimal solutions (Joshi et al., 2019; Hudson et al., 2022; Qiu et al., 2022; Sun & Yang, 2023; Min et al., 2023; Yu et al., 2024). With the learned heatmap, these solvers can greatly reduce the search space. Despite showing better scalability, NAR solvers often depend on post-search procedures, which can be either time-consuming or ineffective in handling VRP constraints, even for the simple cases such as CVRP. Furthermore, recent research has expanded to multi-objective optimization (Fan et al., 2025) and LLM-based approaches (Shi et al., 2026).

2) *Improvement-based solvers* learns to iteratively improve initial solutions, drawing inspiration from classic (meta-)heuristics such as k-opt (e.g., 2-opt (Costa et al., 2020; Wu et al., 2021; Ma et al., 2021), extended to flexible *k*-opt (Ma et al., 2023)), ruin-and-repair (Hottung & Tierney, 2022; Ma et al., 2022), and crossover (Kim et al., 2023). In general, improvement-based methods can achieve near-optimal solutions given prolonged search time, whereas construction-based methods typically offer a more efficient trade-off between performance and runtime.

## B    Problem selection and data generation

As a single-task solver focused on constraint handling, *our work aligns with prior single-task studies that typically evaluate 2-3 representative VRPs.* Specifically, we consider three representative VRPs: complex constrained VRPs where feasibility masking is NP-hard thus intractable (TSPTW) or tractable but ineffective (CVRPBLTW), and simpler constraints (CVRP) where feasibility masking is tractable and effective. Below we introduce the detailed data generation process. Following the convention, all the node coordinates are generated under a uniform distribution, i.e., $x_i, y_i \sim \mathcal{U}[0, 1]$.

### B.1    Traveling Salesman Problem with Time Window (TSPTW)

We primarily follow the settings of a recent study (Bi et al., 2024), which introduced TSPTW with three difficulty levels: Easy (Chen et al., 2024), Medium, and Hard. Given our focus on complex constrained VRPs, we report results on the *Hard* variant in the main tables, consistent with the benchmark dataset in (Da Silva & Urrutia, 2010). Except for the experiment in Table 8, where we use the Medium variant to show how CaR's unified encoder works as constraint hardness changes, all other TSPTW experiments default to the Hard setting.

**NP-hardness of computing feasibility masking on TSPTW.** During solution construction, neural solvers typically apply feasibility masking to exclude actions that violate constraints. In TSPTW, for instance, a node $v_j$ is masked out if its arrival time $t_j$ exceeds its time window end $u_j$, i.e., $t_j > u_j$. However, as highlighted in PIP (Bi et al., 2024), feasibility in TSPTW is not purely local: selecting a node affects the current time, which in turn influences the feasibility of all future selections due

to interdependent time window constraints. For example, a node with a late time window might be locally feasible, but choosing it can delay the tour such that earlier nodes become unreachable, leading to irreversible infeasibility. Ensuring global feasibility thus requires evaluating whether a current decision allows for any feasible completions, i.e., a process that involves simulating all future possibilities. This renders feasibility masking itself NP-hard, compounding the inherent difficulty of solving TSPTW.

**TSPTW Hard**. We first generate a random permutation $\tau$ of the node set and sequentially assign time windows to ensure the existence of a feasible solution for each instance. The time windows are drawn from a uniform distribution, where the lower and upper bounds are given by $l_i \sim \mathcal{U}\left[C_L(\tau'_i) - \eta, C_L(\tau'_i)\right], u_i \sim \mathcal{U}\left[C_L(\tau'_i), C_L(\tau'_i) + \eta\right]$, where $C_L(\tau'_i)$ denotes the cumulative tour length of the partial sequence $\tau'$ up to step $i$, and $\eta$ controls the time window width. To adaptively scale time windows with problem size, we set $\eta = n$, offering greater flexibility than the fixed value of 50 used in (Bi et al., 2024). Concretely, in PIP (Bi et al., 2024), the time window width for TSPTW-Hard decreases as the problem size increases. To scale time windows adaptively with instance size, we set instead of using PIP's fixed value of 50, resulting in a different setting, i.e., a looser time window for TSPTW-100. We note that we evaluate our CaR trained on this new adaptive setting in most of the tables in this work, except Table 10, which uses the original TSPTW-100 dataset from PIP (with significantly tighter time window constraints than our training set). Time windows are then normalized following (Kool et al., 2018) to facilitate neural network training. Since TSP solutions form a Hamiltonian cycle, it is equivalent to setting any node as the starting point. Thus, we designate a specific starting node for TSPTW by redefining its upper bound $u_0$ as $u_0 = \max{(u_i + C_L(e(v_i, v_0)))}, i \in [1, n]$.

**TSPTW Medium**. We generate the lower and upper bounds of the time windows following a uniform distribution: $l_i \sim \mathcal{U}[0, T_N]$, where $T_N$ estimates the expected tour length for the given problem scale (e.g., $T_{20} \approx 10.9$ (Chen et al., 2024)). The upper bound $u_i$ is derived from $l_i$ as $u_i \sim l_i + T_N \cdot \mathcal{U}[0.1, 0.2]$. While this data generation rule does not guarantee instance feasibility, preliminary results show that the time windows overlap significantly, leading to a high feasibility rate in the generated instances.

For all the TSPTW instances, we normalize all $l_i$ and $u_i$ by $u_0$ to ensure their values fall within $[0, 1]$. Training data is generated on the fly, while inference uses the dataset from (Bi et al., 2024) for fair comparison. As noted in (Bi et al., 2024), all test instances are verified to be feasible, either empirically or theoretically.

### B.2 Capacitated VRP with Backhaul, Duration Limit, and Time Window Constraints (CVRPBLTW)

CVRPBLTW follows the setting in (Zhou et al., 2024; Liu et al., 2024; Li et al., 2021) and includes four key constraints: 1) *Capacity (C)*. Each node's demand $q_i$ is sampled from $\mathcal{U}(1, \ldots, 9)$, and the vehicle's capacity $Q$ varies by problem scale, with $Q^{50} = 40$ and $Q^{100} = 50$. 2) *Backhaul (B)*. Node demands are sampled from a discrete uniform distribution $\{1, \ldots, 9\}$, with 20% of customers randomly designated as backhauls. Routes include both linehauls and backhauls without strict precedence constraints. 3) *Duration Limit (L)*. The maximum route length is set to $\ell = 3$, ensuring feasible solutions in the unit square space $\mathcal{U}(0, 1)$. 4) *Time Window (TW)*. The depot time window is defined as $[l_0, u_0] = [0, 3]$, and each customer node has a service time of $s_i = 0.2$. The time window for node $v_i$ is computed as follows: the center is sampled as $\gamma_i \sim \mathcal{U}(l_0 + C_L(e(v_0, v_i)), u_0 - C_L(e(v_0, v_i)) - s_i)$, where $C_L(e(v_0, v_i)) = C_L(e(v_i, v_0))$ represents the travel time between the depot $v_0$ and node $v_i$. The half-width is drawn from $w_i \sim \mathcal{U}(\frac{s_i}{2}, \frac{u_0}{3}) = \mathcal{U}(0.1, 1)$. Finally, the time window is set as $[l_i, u_i] = [\max(l_0, \gamma_i - w_i), \min(u_0, \gamma_i + w_i)]$. Training data is generated on the fly, while inference uses the 1k-instance dataset from (Zhou et al., 2024) for fair comparison.

### B.3 Capacitated Vehicle Routing Problem (CVRP)

We follow the standard setting (Kool et al., 2018; Ma et al., 2021; Kwon et al., 2020). Each node's demand $q_i$ is sampled from $\mathcal{U}(1, \ldots, 9)$, with vehicle capacities set to $Q^{50} = 40$ and $Q^{100} = 50$. Training data is generated on the fly, while inference uses the dataset from (Zhou et al., 2024) for fair comparison.

### B.4 Traveling Salesman Problem with Draft Limit (TSPDL)

TSPDL arises in marine transportation, where vessel capacity constraints must be considered. Each node corresponds to a port with a demand $q_i$ and a draft limit $d_i$. Instances are generated from a TSP by assigning $\sigma\%$ of nodes draft limits smaller than the total demand, i.e., $d_i \sim [q_i, \sum_{j=0}^{n} q_j]$, while the remaining nodes are set to $d_i = \sum_{j=0}^{n} q_j$. Feasible solutions are guaranteed during instance generation. Following Bi et al. (2024), we adopt the Hard setting with $\sigma = 90$, and during inference we directly use the datasets from Bi et al. (2024) for fair comparison.

### B.5 Sequential Ordering Problem (SOP)

SOP (Montemanni et al., 2008) is a routing problem defined by precedence constraints. It can be defined on a graph $\mathcal{G} = \{\mathcal{V}, \mathcal{E}\}$. The objective is to find a permutation $\tau = \{v_1, \cdots, v_n\}$ with fixed start and end nodes $v_1$ and $v_n$ that minimizes the total travel cost, subject to all precedence constraints $(v_i, v_j) \in \mathcal{P}$, i.e., $v_i$ must precede $v_j$. To ensure feasibility, we sample a random valid permutation, identifying all implied precedence pairs. We then randomly sample $h\%$ of these pairs as constraints using a mixture of pairwise Euclidean distance (weight $g$) and random noise (weight $1 - g$). We evaluate CaR on two SOP variants: Variant 1 ($h = 20$, $g = 0.3$) and Variant 2 ($h = 20$, $g = 0.8$).

## C Network architecture

The network architectures of mainstream neural construction and improvement solvers are typically based on a Transformer encoder with an attention-based decoder, enabling unification across both paradigms. In this paper, we adopt a unified encoder shared by the construction and refinement decoders. Specifically, we use a 6-layer Transformer encoder, following POMO (Kwon et al., 2020), while retaining the original decoders from NeuOpt (Ma et al., 2023) and N2S (Ma et al., 2022), corresponding to two representative local search operators: flexible $k$-opt (Ma et al., 2023) and remove-and-reinsertion (R&R) (Ma et al., 2022), respectively. For the unexplored variants TSPTW and CVRPBLTW, we design new constraint-related features analogous to the contextual features used in the original decoders. The concrete forward processes are introduced below.

### C.1 Encoder

As shown in Figure 2, the construction module first takes node features $f_i^n$—including coordinates $(x_i, y_i)$ and constraint-related features (e.g., time windows $[l_i, u_i]$ and demand $q_i$)—as input. For each node $v_i$, these features are projected into a $d$-dimensional embedding $h_i^{(0)} \in \mathbb{R}^d$ ($d = 128$) via a linear layer. The initial embedding is then passed through a 6-layer Transformer network (Kwon et al., 2020). At each layer $j$ ($j = 1, \cdots, 6$), the embedding $h_i^{(j-1)}$ is projected into query, key, and value vectors:

$$q_i^{(j)} = W_q^{(j)} h_i^{(j-1)}, \quad k_i^{(j)} = W_k^{(j)} h_i^{(j-1)}, \quad v_i^{(j)} = W_v^{(j)} h_i^{(j-1)}, \tag{6}$$

where $W_q^{(j)}, W_k^{(j)}, W_v^{(j)} \in \mathbb{R}^{d \times d}$. These are fed into a multi-head attention (MHA) layer, whose output is:

$$\tilde{h}_i^{(j)} = \text{Softmax}\left(\frac{(q_i^{(j)})^\top k_i^{(j)}}{\sqrt{d}}\right) v_i^{(j)}. \tag{7}$$

The MHA output is then linearly transformed:

$$\hat{h}i^{(j)} = W_{\text{MHA}}^{(j)} \tilde{h}i^{(j)}, \tag{8}$$

where $W\text{MHA}^{(j)} \in \mathbb{R}^{d \times d}$. This passes through instance normalization with residual connection:

$$h'^{(j)}_i = \text{IN}\left(\hat{h}_i^{(j)} + h_i^{(j-1)}\right). \tag{9}$$

Next, a feed-forward (FF) layer refines the output:

$$h''^{(j)}_i = W_2^{(j)} \cdot \text{ReLU}\left(W_1^{(j)} h'^{(j)}_i\right), \tag{10}$$

where $W_1^{(j)} \in \mathbb{R}^{d \times d'}$, $W_2^{(j)} \in \mathbb{R}^{d' \times d}$, and $d' = 512$ as in (Kwon et al., 2020). The final embedding at layer $j$ is:

$$h_i^{(j)} = \text{IN}\left(h''^{(j)}_i + h'^{(j)}_i\right). \tag{11}$$

Thereafter, this process is repeated for 6 layers, and the final encoder output is $\boldsymbol{h}_i = \boldsymbol{h}_i^{(6)}$.

When it comes to the refinement module, the input changes from a node feature set to a linked list (i.e., the solution), which consists of the original node features plus positional information and a pointer to the next node in the solution. Notably, for CVRP variants, the depot node may appear multiple times, with each occurrence treated as a distinct node due to its different position. The refinement module shares the same operations as the construction module, except for Eq. (7). Specifically, we first incorporate the cyclic positional encoding (CPE) from (Ma et al., 2021) to obtain the positional embedding $p_i \in \mathbb{R}^d$. We then apply a self-attention mechanism:

$$\tilde{p}_i = (W_q^p p_i)^\top (W_k^p p_i), \tag{12}$$

where $W_q^p, W_k^p \in \mathbb{R}^{d \times d}$, to compute the positional attention matrix $a^s$. We replace Eq. (7) from the construction module with:

$$\tilde{\boldsymbol{h}}_i^{(j)} = \text{Softmax}\left(\frac{\text{MLP}\left(\left[(\boldsymbol{q}_i^{(j)})^\top \boldsymbol{k}_i^{(j)}, \tilde{p}_i\right]\right)}{\sqrt{d}}\right) \boldsymbol{v}_i^{(j)}, \tag{13}$$

where the MLP reduces the dimension of the concatenated attention scores, i.e., $[a^n, a^s]$ in Figure 2, from 2 to 1, following the Syn-Att mechanism proposed in (Ma et al., 2022).

## C.2 DECODER

*The construction decoder* primarily employs an MHA layer, where the key and value vectors are linearly transformed from the node embedding $h_i$. The query vector is computed using contextual information from the construction process, including: 1) the embedding of the partial solution $h^s$ (i.e., the node embedding of the last node in the partial solution), and 2) the step-wise solution features $f^s$, which include the vehicle's remaining load, current time, and current sub-tour length. The MHA output is calculated as:

$$a = \sum_{i=0}^{n} \text{Softmax}\left(\frac{q_i^\top k_i}{\sqrt{d}} + \xi_i\right) v_i, \tag{14}$$

where $\xi_i$ is the feasibility mask for node $v_i$. We mask out visited nodes to ensure solution validity across all variants. For CVRP, as discussed in Appendix E.1, we also mask nodes that violate capacity constraints at each construction step. The output of each head is then passed through a linear layer parameterized by $W_o$, yielding $h_a = W_o a$. Finally, the decoder computes selection probabilities for all candidate nodes using a single-head attention layer:

$$p_i = \text{Softmax}\left(\zeta \cdot \tanh\left(\frac{h_a^\top h_i}{\sqrt{d}}\right) + \xi_i\right), \tag{15}$$

where $\zeta$ scales the logits to encourage exploration (Kwon et al., 2020).

*The refinement decoder* follows the original design in the NeuOpt (Ma et al., 2023) and N2S (Ma et al., 2022), which performs two representative local search operators: the flexible $k$-opt[1] and remove-and-reinsertion (R&R), respectively. Figure 5 illustrates the detailed improvement process in a single time step for different local search operators, where each node selection corresponds to a full computation of the network decoder. Note that although the original NeuOpt and N2S decoders adopt different architectures, their input and output formats are similar. At each refinement step $t = 1, \ldots, T_R$, the inputs include: (i) the updated *node embedding* $h_i^t$ derived from the previous solution $\tau_{t-1}$ (with $\tau_0$ as the top-$p$ initial construction solutions); (ii) *refinement features* encoding feasibility transition history for $k$-opt or removal action history for R&R; and (iii) *refinement embedding* of the last action node, analogous to the solution embedding in construction. The output is the selection probability over candidate nodes for the next refinement operation (e.g., $k$-opt, removal, or insertion).

To adapt improvement solvers to new variants (e.g., TSPTW, CVRPBLTW), we follow their original design principles while integrating variant-specific features, such as refinement history and node-level feasibility information. Specifically, when using the NeuOpt decoder, we encode the feasibility of the most recent three refinement steps as a binary vector to represent refinement history and incorporate node-level feasibility features to enhance constraint awareness. For TSPTW, these features include

---

[1]Following (Ma et al., 2023), we set $k \leq 4$.

the arrival time at each node, time window violation value, last node arrival time, and an indicator of whether the solution becomes infeasible after visiting the current node. For CVRPBLTW, node-level feasibility features include binary indicators for depot and backhaul nodes, violation values for each constraint, constraint-related attributes (e.g., arrival time and cumulative distance/demand), and infeasibility markers indicating whether the solution becomes infeasible before or after visiting the current node. Adapted NeuOpt variants are marked with ‡ in the main tables.

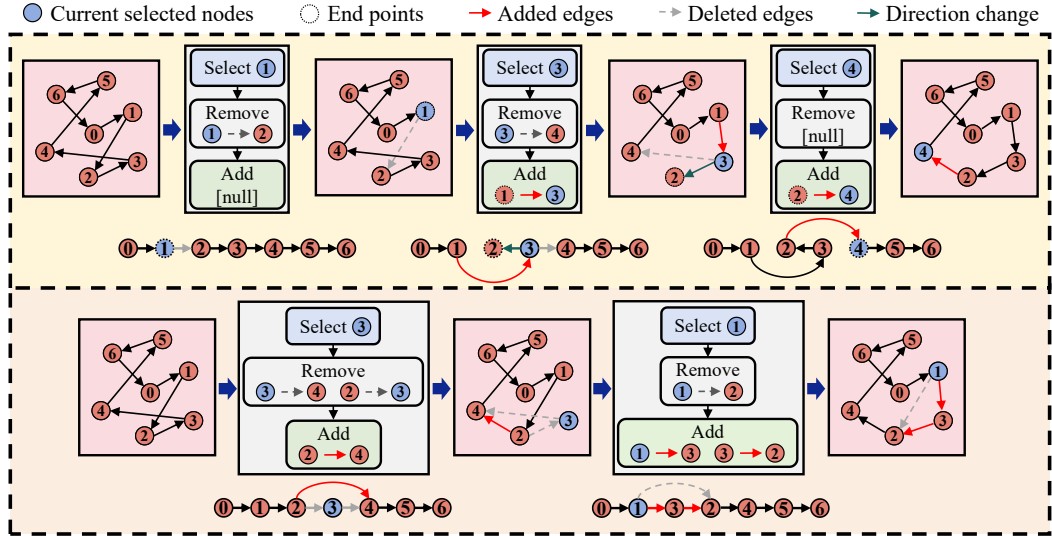

Figure 5: Illustration of actions and operations in *k*-opt (*top*) and remove-and-reinsertion (*bottom*).

## D  IMPLEMENTATION DETAILS

### D.1  EXPERIMENTAL SETTINGS

We preset refinement steps $T_R = 5$ during training. Models are trained using Adam with a learning rate of $1 \times 10^{-4}$, weight decay of $1 \times 10^{-6}$. Follow the setting of (Kwon et al., 2020), we use instance normalization in the MHA and set the embedding dimension to 128. To prevent out-of-memory issues early in training, refinement is activated after a ~10-epoch warmup of the construction module. To optimize GPU memory utilization, we adjust the number of refined solutions $p$ separately for each problem size during training. We set $\alpha_1 = 0.01$ and $\alpha_2 = 1$ in the construction loss, and set $\omega = 100$ for TSPTW and CVRP, and 10 for CVRPBLTW based on the relative loss scales between modules observed during warm-up epochs. During inference, TSPTW and CVRPBLTW results are obtained without multi-starting, consistent with the training setup. For each augmented instance, we sample one solution for TSPTW and greedily generate one for CVRPBLTW ($p = 1$), which are then passed to the refinement module. For CVRP, Table 12 shows that multi-starting improves performance, so we apply it during construction but refine only the top $p$ solutions for efficiency, setting $p = 2$ for $n = 50$ and $p = 1$ for $n = 100$ due to the GPU memory constraint. We use only one GPU for all problem sizes during inference to ensure fair comparison across variants.

### D.2  BASELINE

We compare our CaR framework with state-of-the-art classic and neural VRP solvers. Since CaR is positioned as a single-task solver, we primarily compare it against the best-performing single-task, single-paradigm methods, namely, construction, improvement, and post-construction search approaches[2], for the unexplored hard-constrained problems TSPTW and CVRPBLTW. For com-

---

[2]We do not consider post-construction search methods (e.g., fine-tuning via EAS (Hottung et al., 2022), beam search via SGBS (Choo et al., 2022), and random reconstruction via RRC (Luo et al., 2023)) as cross-paradigm like CaR, since they rely on the construction policy. These approaches face inherent limitations for constraint handling.

pleteness, we also include comparisons with recent multi-task solvers, particularly on benchmark problems like CVRP and CVRPBLTW. Below, we detail the baseline selection for each problem.

1) *TSPTW*: We primarily compare against PIP, the *SoTA* construction solver. No improvement or post-search methods have been implemented on TSPTW in prior work, so we adapt the single-task *SoTA* improvement solver NeuOpt (Ma et al., 2023) (denoted as NeuOpt*‡) and the strongest post-search method UDC+RRC (Zheng et al., 2024) by incorporating our Lagrangian-relaxed reward and constraint-related features. This allows us to highlight CaR's cross-paradigm advantage over all types of single-paradigm methods. Multi-task solvers are excluded, as they rely on feasibility masking, which is intractable for TSPTW.

2) *CVRPBLTW*: As this variant has only been addressed by multi-task solvers in prior work, we primarily compare against them. Among single-task methods, POMO (Kwon et al., 2020), as implemented in MVMoE (Zhou et al., 2024), is the best-performing construction solver. No single-task improvement or post-search methods have been applied to CVRPBLTW, so we adapt the *SoTA* improvement solver NeuOpt (Ma et al., 2023) (denoted as NeuOpt*‡) and the strongest post-search method UDC+RRC (Zheng et al., 2024) by incorporating our Lagrangian-relaxed reward and constraint-related features. However, UDC cannot handle multiple constraints simultaneously, so we instead adapt the relaxed reward to SGBS (Choo et al., 2022) on top of POMO.

3) *CVRP*: CVRP is a well-studied variant that most of the neural VRP solvers have implemented on. For comprehensiveness, we compare with multiple recent baselines listed in Table 11.

Table 11: Neural baselines for CVRP result comparison.

| Type | Paradigm | Neural baselines |
|---|---|---|
| Single-task | Construction | AM (Kool et al., 2018), POMO (Kwon et al., 2020), BQ-NCO (Drakulic et al., 2023), LEHD (Luo et al., 2023), UDC (Zheng et al., 2024), InViT (Fang et al., 2024), PolyNet (Hottung et al., 2025) |
| Single-task | Construction + Post-search | SGBS (Choo et al., 2022) (on top of POMO+EAS (Hottung et al., 2022)), RRC (Luo et al., 2023) (on top of LEHD and UDC) |
| Multi-task | Construction | POMO-MTL (Liu et al., 2024), MVMoE (Zhou et al., 2024), ReLD-MoEL (Huang et al., 2025) |
| Single-task | Improvement | NeuOpt-GIRE (Ma et al., 2023) |
| Single-task | Construction + Improvement | AM+LCP (Kim et al., 2021) |

We then introduce the implementation details for each baseline.

1) Classic solvers:

- *LKH-3* (Helsgaun, 2017), a strong solver capable of handling multiple VRP variants, run with a maximum of 10,000 trials and 1 run as in (Kool et al., 2018). We apply LKH-3 to CVRP and TSPTW; however, it is not used for CVRPBLTW due to lack of support.

- *OR-Tools* (Furnon & Perron, 2024), an open-source combinatorial optimization suite, where we use cheapest insertion strategies for initialization and guided local search under time limits of 20/40 (short) and 200/400 (long) seconds for $n = 50/100$, following (Zhou et al., 2024; Bi et al., 2024). We apply it to TSPTW and CVRPBLTW due to the absence of strong classic solvers for these variants.

- *Greedy Heuristic*, a hand-crafted heuristic selecting the nearest node (Greedy-L) or the node with the soonest time window end (Greedy-C) during construction. We apply it only to TSPTW following (Bi et al., 2024).

- *HGS* (Vidal et al., 2012), a state-of-the-art heuristic solver for CVRP. We directly cite the results of HGS on CVRP and CVRPBLTW from MVMoE (Zhou et al., 2024), as we use the same test dataset.

2) Neural construction solvers:

- *AM* (Kool et al., 2018), a classical neural construction method that first introduced the Transformer architecture for solving various VRP variants via reinforcement learning. We evaluate it only on CVRP (Table 14) due to its inferior performance on constrained VRPs and page limits. Pretrained models are used for evaluation.

- *POMO* (Kwon et al., 2020), an enhanced version of AM, widely adapted for multiple VRPs—CVRP (Kwon et al., 2020), TSPTW (Bi et al., 2024), and CVRPBLTW (Zhou et al., 2024). We retrain POMO on all variants to match CaR's gradient steps for fair comparison.

- *BQ-NCO* (Drakulic et al., 2023), a generic framework that reformulates the MDP using bisimulation quotienting to reduce the state space and exploit problem symmetries. We report its greedy decoding results on CVRPLIB for comparison.
- *LEHD* (Luo et al., 2023), a novel framework that rethinks conventional architectures by adopting a light encoder and heavy decoder design, trained via supervised learning. We evaluate it only on CVRP, as extending it to new domains would require extensive label generation.
- *UDC* (Zheng et al., 2024), a recent unified divide-and-conquer framework designed for large-scale VRPs. For TSPTW, we retrain UDC with our relaxed objective (Table 2). For CVRPBLTW, applying relaxed CMDP is impractical due to the difficulty in defining a consistent objective across sub-tours under multiple constraints, which goes beyond UDC's intended design. For CVRP, we use the released pretrained model trained on $n = 500$–$1000$, adapting sub-tour lengths to 25 for $n = 50$ and 50 for $n = 100$ (Table 14).
- *InViT* (Fang et al., 2024), a generalizable solver employing a nested-view Transformer encoder to capture multi-scale local structures while enforcing permutation invariance. We compare with its pretrained model on CVRP.
- *PolyNet* (Hottung et al., 2025), a fine-tuning method that introduces random vectors into the decoder to enhance solution diversity. Due to the unavailability of source code, we directly cite its performance on CVRP from the original paper for comparison.
- *EAS* (Hottung et al., 2022), a fine-tuning approach that adapts pretrained model parameters per instance. We use the POMO+EAS version as the base model of SGBS for comparison.
- *SGBS* (Choo et al., 2022), or simulation-guided beam search, integrates neural construction with simulation rollouts. For CVRPBLTW, we adapt it with the relaxed reward (*); for CVRP, we use the released pretrained model. As the default 28-iteration setup is time-consuming, we also report a single-iteration version (*short*) for fair comparison with CaR.
- *POMO-MTL* (Liu et al., 2024), a multi-task solver designed to handle multiple VRP variants with a single model. We directly cite its results on CVRP and CVRPBLTW from MVMoE (Zhou et al., 2024), as we use the same test dataset.
- *MVMoE* (Zhou et al., 2024), a multi-task solver enhanced by a mixture-of-experts mechanism and implemented on 16 VRP variants. We evaluate it using the released pretrained models.
- *ReLD-MoEL* (Huang et al., 2025), a multi-task solver that enriches contextual information during auto-regressive solution construction. Due to the unavailability of code, we directly cite its reported performance on CVRP and CVRPBLTW from the original paper for comparison.
- *PIP* (Bi et al., 2024), a novel framework that learns approximate feasibility masking for complex VRPs with interdependent constraints, such as TSPTW. We retrain the model under our settings to ensure fair comparison.

3) Neural improvement solvers:

- *NeuOpt-GIRE* (Ma et al., 2023), the *SoTA* single-task improvement method that uses neural networks to perform flexible $k$-opt operations. GIRE explores infeasible regions to enhance performance. We follow its default settings for training and testing. For CVRP, we directly evaluate using its pretrained model. For TSPTW and CVRPBLTW, which are not implemented in the original paper, we adapt the method with the relaxed reward function (*) and our tailored constraint-related features (‡), denoted as NeuOpt-GIRE*‡. For all variants, we report results with 1k, 2k, and 5k iterations using random initial solutions.

4) Neural hybrid solvers:

- *LCP* (Kim et al., 2021), a single-task cross-paradigm method where the seeder constructs initial solutions, decomposes them into sub-tours, and the reviser independently refines and merges them into complete solutions. Due to the unavailability of source code, we directly cite its reported performance on top of AM for CVRP from the original paper.
- *NCS* (Kong et al., 2024), a single-task cross-paradigm method where the improvement policy is trained with a construction policy via a shared critic. It is implemented on the pickup and delivery problems (PDP) and still rely on heavy improvement. To maximize comparison, we adapt it to TSPTW using our constraint-related features and penalty guided.

# E ADDITIONAL EXPERIMENTS AND DISCUSSION

## E.1 DISCUSSION OF THE EXISTING FEASIBILITY MASKING

We now present our insights into the limitations of existing feasibility masking mechanisms. As shown in Table 12, removing feasibility masking for CVRP hampers its performance, in contrast to more complex problems like CVRPBLTW, where its removal leads to improved results (see results of POMO* vs. POMO in Table 2). To further investigate this, we compare the two model variants with and without masking on CVRPBLTW-100. Figure 7 shows that the mode with strict feasibility masking (red) converges rapidly in terms of training loss (saturating after 2k epochs) but yields noticeably lower-quality solutions, while the one without masking (blue) converges more gradually but ultimately attains significantly better solution quality. This finding is consistent with prior works (Mani et al., 2025; Zhang et al., 2023), which shows that excessive action-space pruning leads to conservative policy behavior and degraded learning performance These results suggest that *while existing feasibility masking is sufficient for simple problems, it is far from a silver bullet for handling complex constraints.* In fact, for highly constrained problems, feasibility masking is often intractable (e.g., TSPTW) or ineffective (e.g., CVRPBLTW), resulting in infeasibility and sub-optimality.

Table 12: Effects of the multi-starting and feasibility masking during construction on

| *w.* multi-start | *w.* mask | Obj.↓ | Gap↓ | Infsb%↓ |
|:---:|:---:|:---:|:---:|:---:|
| ✗ | ✗ | 10.550 | 2.093% | 0.00% |
| ✗ | ✓ | 10.520 | 1.793% | 0.00% |
| ✓ | ✗ | 10.431 | 0.921% | 0.00% |
| ✓ | ✓ | 10.424 | **0.857%** | 0.00% |

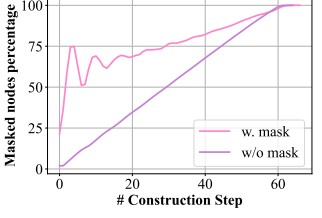 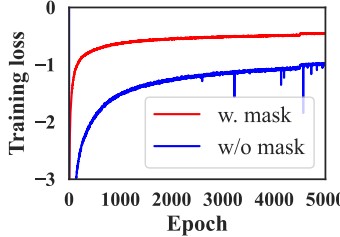 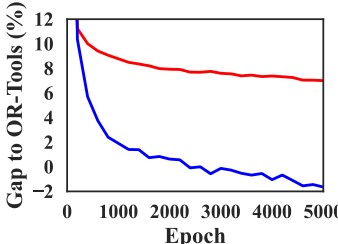

Figure 6: Masked node count in CVRPBLTW construction.

Figure 7: Curve of training loss (left) and optimality gap (right).

To address these challenges, we propose the CaR framework. For TSPTW, CaR leverages refinement to correct infeasibility and improve solution quality. For CVRPBLTW, we found that removing this restrictive mask significantly improves the performance (Gap: 9.17% vs. 2.31%), but it leads to unexpected infeasibility (2.6%). CaR addresses this by applying refinement, which further reduces infeasibility and improves solution quality. However, a small fraction of infeasible solutions can still remain. Since feasibility can ultimately be guaranteed through masking, we apply a final reconstruction step with feasibility masking during inference to ensure all reported solutions are valid (see Table 13). This strategy allows CaR to relax constraints during construction and refinement to enhance quality while maintaining overall feasibility through targeted post-processing. Note that in this paper, "without masks" does not imply removing all masks. All VRP variants must satisfy the fundamental constraint that each node (except depots) is visited exactly once. To ensure the validity of the generated solutions, we retain the masking of visited nodes during the construction process.

## E.2 RESULTS ON CVRP

To assess CaR's extendability, we evaluate it on the simple CVRP, where feasibility masking is tractable and effective. In this setting, CaR shifts focus toward optimizing solution quality while still benefiting from limited exploration of infeasible regions. Table 14 compares CaR with classic, single-paradigm, and cross-paradigm methods. CaR delivers strong performance within a short

Table 13: CaR module impact on CVRPBLTW feasibility and quality (**Bold**: reported in Table 2).

| Method | Module | n=50 Obj.↓ | Gap↓ | Infsb%↓ | n=100 Obj.↓ | Gap↓ | Infsb%↓ |
|---|---|---|---|---|---|---|---|
| CaR | Construction (w/o mask) | 14.798 | 2.116% | 3.50% | 24.404 | -2.050% | 2.80% |
| | Refinement (k-opt) | 14.759 | 1.682% | 2.50% | 24.360 | -2.285% | 2.60% |
| | Re-construction (w. mask) | **14.844** | **2.114%** | 0.00% | **24.585** | **-1.724%** | 0.00% |
| CaR | Construction (w/o mask) | 14.843 | 2.235% | 2.60% | 24.460 | -1.715% | 3.10% |
| | Refinement (R&R) | 14.567 | 0.221% | 1.10% | 24.215 | -2.886% | 1.90% |
| | Re-construction (w. mask) | **14.601** | **0.463%** | 0.00% | **24.400** | **-2.448%** | 0.00% |

inference time, for example, achieving a 0.889% optimality gap on CVRP-100 in 20s using only 20 refinement steps, whereas NeuOpt-GIRE requires over 5 minutes with 5,000 steps. When combined with EAS, CaR further outperforms all baselines, achieving the *SoTA* results.

We validate CaR's generalization under distributional shift on the real-world CVRPLIB dataset (Uchoa et al., 2017), as shown in Figure 4. Since 1) extensive post-search affects the evaluation of generalization performance, and 2) prolonged search is impractical for deployment, we test all baselines using a greedy strategy for fair comparison. CaR likewise uses $T_R = 20$ with a greedy strategy, consistent with Table 14. POMO is retrained under the same conditions (epochs and batch size) as CaR. As shown in Table 15, CaR achieves a gap of 5.00%, outperforming the best prior neural method MvMoE (Zhou et al., 2024) with the optimality gap: 6.88%, showing that CaR's *extendability* on simple constrained CVRP and good generalization performance.

Table 14: Results on CVRP: best are bolded; best within 1 min / 2min ($n = 50/100$) shaded.

| Method | #Params | Paradigm§ | n = 50 Obj.↓ | Gap↓ | Time | n = 100 Obj.↓ | Gap↓ | Time |
|---|---|---|---|---|---|---|---|---|
| HGS | / | I | 10.334 | ◇ | 4.6m | 15.504 | ◇ | 9.1m |
| LKH-3 | / | I | 10.346 | 0.115% | 9.9m | 15.590 | 0.556% | 18.0m |
| OR-Tools (short) | / | I | 10.540 | 1.962% | 10.4m | 16.381 | 5.652% | 20.8m |
| OR-Tools (long) | / | I | 10.418 | 0.788% | 1.7h | 15.935 | 2.751% | 3.5h |
| AM | 0.68M | L2C-S | 10.590 | 2.478% | 17s | 16.139 | 4.098% | 1.1m |
| POMO | 1.25M | L2C-S | 10.424 | 0.867% | 1s | 15.741 | 1.532% | 5s |
| LEHD (greedy) | 1.42M | L2C-S | 10.861 | 5.102% | 1.2s | 16.148 | 4.154% | 2s |
| InViT | 1.74M | L2C-S | 10.827 | 4.773% | 6.3m | 16.452 | 6.117% | 9.6m |
| PolyNet # | / | L2C-S | / | / | / | 15.640 | 0.490% | 5m |
| POMO-MTL | 1.25M | L2C-M | 10.437 | 0.987% | 2s | 15.790 | 1.846% | 7s |
| MVMoE | 3.68M | L2C-M | 10.428 | 0.896% | 3s | 15.760 | 1.653% | 10s |
| ReLD-MoEL | 3.68M | L2C-M | 10.425 | 0.866% | 3s | 15.713 | 1.354% | 9s |
| POMO+EAS+SGBS (short) | 1.25M | L2C-S | 10.364 | 0.293% | 15s | 15.619 | 0.744% | 1.3m |
| POMO+EAS+SGBS (long) | 1.25M | L2C-S | 10.340 | 0.056% | 6.5m | 15.530 | 0.170% | 34m |
| LEHD (RRC=50) | 1.42M | L2C-S | 10.419 | 0.821% | 28s | 15.674 | 1.096% | 36s |
| LEHD (RRC=500) | 1.42M | L2C-S | 10.367 | 0.315% | 3.5m | 15.581 | 0.497% | 6.5m |
| UDC (RRC=50) † | 1.50M | L2C-S | 10.722 | 3.755% | 2.8m | 16.212 | 4.567% | 9.8m |
| UDC (RRC=250) † | 1.50M | L2C-S | 10.639 | 2.953% | 12.2m | 16.038 | 3.442% | 46m |
| NeuOpt-GIRE ($T = 1k$) | 0.69M | L2I-S | 10.411 | 0.749% | 33s | 15.809 | 1.969% | 1.4m |
| NeuOpt-GIRE ($T = 2k$) | 0.69M | L2I-S | 10.377 | 0.414% | 1.1m | 15.724 | 1.417% | 2.7m |
| NeuOpt-GIRE ($T = 5k$) | 0.69M | L2I-S | 10.355 | 0.203% | 2.8m | 15.640 | 0.878% | 6.8m |
| AM+LCP # | / | L2C+L2I-S | 10.520 | 1.380% | 5.2m | 15.980 | 2.110% | 29m |
| CaR ($T_R = 5$) | 1.64M | L2C+L2I-S | 10.372 | 0.360% | 4s | 15.673 | 1.085% | 9s |
| CaR ($T_R = 10$) | 1.64M | L2C+L2I-S | 10.366 | 0.305% | 7s | 15.651 | 0.943% | 14s |
| CaR ($T_R = 20$) | 1.64M | L2C+L2I-S | 10.362 | 0.259% | 13s | 15.642 | 0.889% | 23s |
| CaR (sample 4, $T_R = 20$) | 1.64M | L2C+L2I-S | 10.350 | 0.150% | 54s | 15.604 | 0.639% | 1.7m |
| CaR (sample 16, $T_R = 20$) | 1.64M | L2C+L2I-S | 10.344 | 0.096% | 3.6m | 15.578 | 0.476% | 6.6m |
| CaR ($T_R = 20$) + EAS | 1.64M | L2C+L2I-S | 10.341 | 0.063% | 7.8m | 15.540 | 0.230% | 25m |
| CaR (sample 4, $T_R = 20$) + EAS | 1.64M | L2C+L2I-S | **10.338** | **0.034%** | 31m | **15.527** | **0.148%** | 1.7h |

# Due to the unavailability of the source code, we directly copy their results from the original paper for comparison.

### E.3 QUANTIFICATION FOR THE DIVERSITY OF THE CONSTRUCTED INITIAL SOLUTIONS

We now provide a quantitative analysis across a large set of instances. Specifically, we evaluate the average pairwise diversity among the eight constructed solutions for 10,000 instances, comparing

Table 15: Results on CVRPLIB instances. Best results are bolded.

| Instance | Opt | POMO (greedy) Obj. | Gap | BQ-NCO (greedy) Obj. | Gap | LEHD (greedy) Obj. | Gap | INViT (greedy) Obj. | Gap | UDC (RRC=2) Obj. | Gap | MVMoE (greedy) Obj. | Gap | CaR (greedy, $T_R = 20$) Obj. | Gap |
|---|---|---|---|---|---|---|---|---|---|---|---|---|---|---|---|
| X-n101-k25 | 27591 | 30138 | 9.23% | 33617 | 21.84% | 31445 | 13.97% | 28311 | **2.61%** | 33466 | 21.29% | 29361 | 6.42% | 28518 | 3.36% |
| X-n106-k14 | 26362 | 39322 | 49.16% | 28221 | 7.05% | 27351 | 3.75% | 27614 | 4.75% | 28681 | 8.80% | 27278 | **3.47%** | 27703 | 5.09% |
| X-n110-k13 | 14971 | 15223 | 1.68% | 15718 | 4.99% | 15252 | 1.88% | 15729 | 5.06% | 16665 | 11.32% | 15089 | **0.79%** | 15232 | 1.74% |
| X-n115-k10 | 12747 | 16113 | 26.41% | 15266 | 19.76% | 13950 | 9.44% | 13744 | 7.82% | 16056 | 25.96% | 13847 | 8.63% | 13097 | **2.75%** |
| X-n120-k6 | 13332 | 14085 | 5.65% | 15524 | 16.44% | 13851 | 3.89% | 14363 | 7.73% | 16618 | 24.65% | 14089 | 5.68% | 13483 | **1.13%** |
| X-n125-k30 | 55539 | 58513 | 5.35% | 62820 | 13.11% | 65475 | 17.89% | 59527 | 7.18% | 61745 | 11.17% | 58944 | 6.13% | 57937 | **4.32%** |
| X-n129-k18 | 28940 | 29246 | **1.06%** | 30665 | 5.96% | 30100 | 4.01% | 31038 | 7.25% | 32203 | 11.28% | 29802 | 2.98% | 29299 | 1.24% |
| X-n134-k13 | 10916 | 11302 | 3.54% | 11989 | 9.83% | 11892 | 8.94% | 11524 | 5.57% | 11951 | 9.48% | 11353 | 4.00% | 11232 | **2.89%** |
| X-n139-k10 | 13590 | 14035 | 3.27% | 14843 | 9.22% | 14009 | 3.08% | 14395 | 5.92% | 15336 | 12.85% | 13825 | 1.73% | 13724 | **0.99%** |
| X-n143-k7 | 15700 | 16131 | 2.75% | 17805 | 13.41% | 17900 | 14.01% | 17028 | 8.46% | 18253 | 16.26% | 16125 | 2.71% | 16017 | **2.02%** |
| X-n148-k46 | 43448 | 49328 | 13.53% | 47815 | 10.05% | 60384 | 38.98% | 45342 | 4.36% | 50167 | 15.46% | 46758 | 7.62% | 44603 | **2.66%** |
| X-n153-k22 | 21220 | 32476 | 53.04% | 28481 | 34.22% | 27359 | 28.93% | 23686 | 11.62% | 25621 | 20.74% | 23793 | 12.13% | 23270 | **9.66%** |
| X-n157-k13 | 16876 | 17660 | 4.65% | 18496 | 9.60% | 17673 | 4.72% | 17875 | 5.92% | 18419 | 9.14% | 17650 | **4.59%** | 17720 | 5.00% |
| X-n162-k11 | 14138 | 14889 | 5.31% | 15475 | 9.46% | 14629 | 3.47% | 14958 | 5.80% | 17581 | 24.35% | 14654 | 3.65% | 14580 | **3.13%** |
| X-n167-k10 | 20557 | 21822 | 6.15% | 23334 | 13.51% | 21591 | 5.03% | 21690 | 5.51% | 25806 | 25.53% | 21340 | 3.81% | 21205 | **3.15%** |
| X-n172-k51 | 45607 | 49556 | 8.66% | 52393 | 14.88% | 60785 | 33.28% | 49525 | 8.59% | 53915 | 18.22% | 51292 | 12.47% | 47830 | **4.87%** |
| X-n176-k26 | 47812 | 54197 | 13.35% | 57647 | 20.57% | 60721 | 27.00% | 50131 | **4.85%** | 59378 | 24.19% | 55520 | 16.12% | 51971 | 8.70% |
| X-n181-k23 | 25569 | 37311 | 45.92% | 27052 | 5.80% | 25937 | **1.44%** | 27676 | 8.24% | 28968 | 13.29% | 26258 | 2.69% | 26530 | 3.76% |
| X-n186-k15 | 24145 | 25222 | 4.46% | 26419 | 9.42% | 25024 | 3.64% | 25724 | 6.54% | 29963 | 24.10% | 25182 | 4.29% | 24710 | **2.34%** |
| X-n190-k8 | 16980 | 18315 | 7.86% | 18629 | 9.71% | 17899 | **5.41%** | 18062 | 6.37% | 19756 | 16.35% | 18327 | 7.93% | 18203 | 7.20% |
| X-n195-k51 | 44225 | 49158 | 11.15% | 54286 | 22.75% | 51067 | 15.47% | 48528 | 9.73% | 55152 | 24.71% | 49984 | 13.02% | 46839 | **5.91%** |
| X-n200-k36 | 58578 | 64618 | 10.31% | 65684 | 12.13% | 64588 | 10.26% | 61870 | 5.62% | 63398 | 8.23% | 61530 | **5.04%** | 62754 | 7.13% |
| X-n209-k16 | 30656 | 32212 | 5.08% | 33608 | 9.63% | 31845 | 3.88% | 33084 | 7.92% | 35692 | 16.43% | 32033 | 4.49% | 31801 | **3.73%** |
| X-n219-k73 | 117595 | 133545 | 13.56% | 128214 | 9.03% | 135834 | 15.51% | 120370 | **2.36%** | 128751 | 9.49% | 121046 | 2.93% | 134143 | 14.07% |
| X-n228-k23 | 25742 | 48689 | 89.14% | 31668 | 23.02% | 29451 | 14.41% | 27881 | 8.31% | 31405 | 22.00% | 31054 | 20.64% | 27773 | **7.89%** |
| X-n237-k14 | 27042 | 29893 | 10.54% | 27978 | **3.46%** | 28140 | 4.06% | 30306 | 12.07% | 35192 | 30.14% | 28550 | 5.58% | 28926 | 6.97% |
| X-n247-k50 | 37274 | 56167 | 50.69% | 47644 | 27.82% | 54017 | 44.92% | 42548 | 14.15% | 45746 | 22.73% | 43673 | 17.17% | 42408 | **13.77%** |
| X-n251-k28 | 38684 | 40263 | 4.08% | 40448 | 4.56% | 39601 | **2.37%** | 41547 | 7.40% | 43277 | 11.87% | 41022 | 6.04% | 40406 | 4.45% |
| **Average** | 31280 | 36408 | 16.63% | 35419 | 13.26% | 35992 | 12.27% | 33360 | 7.06% | 36399 | 17.50% | 33549 | 6.88% | 33283 | **5.00%** |

CaR (trained with diversity loss) and PIP (without diversity loss). For each instance, we calculate the following diversity metrics across solution pairs (higher values indicate greater diversity): 1) Hamming Distance (HD), which measures how many positions differ between two solutions, 2) Positional Jaccard Distance (PJD), which measures the proportion of overlapping node positions across all permutations, and 3) Kendall's Tau Distance (KTD), which measures how many node pairs (i.e., edges) have different relative orderings. As shown in Table 16, CaR's diversity-driven sampling strategy significantly improves the diversity of the constructed solutions compared to the *SoTA* constructive baseline (PIP), across multiple diversity metrics.

Table 16: Diversity comparison between PIP and CaR constructed solutions.

| | HD ↑ | PJD ↑ | KTD ↑ |
|---|---|---|---|
| PIP's constructed solutions | 0.666 ± 0.837 | 0.013 ± 0.017 | 0.713 ± 0.938 |
| CaR's constructed solutions | **0.770 ± 0.893** | **0.015 ± 0.018** | **0.821 ± 0.997** |
| CaR's diversity improvement over PIP | +15.6% | +15.6% | +15.1% |

### E.4 DISCUSSION OF SHARED REPRESENTATION

Recall that CaR employs a unified encoder shared across the construction and refinement decoders. As demonstrated in Table 8, this architecture significantly enhances performance. To further assess the impact of the unified encoder under varying constraint complexities, we present comprehensive results in Table 17 for *Medium* TSPTW-50, *Hard* TSPTW-50, and *Medium* TSPTW-100. Results indicate that as problem complexity increases, evidenced by higher infeasibility rates (e.g., 37.270% in *Hard* TSPTW-50) and greater optimality gaps (e.g., 10.931% in TSPTW-100), the performance improvements attributed to the unified encoder become more pronounced. This suggests that shared representations facilitate better generalization and constraint handling in more challenging scenarios.

Table 17: Effects of unified encoder on TSPTW.

| Method | TSPTW-50 Medium Infsb%↓ | Gap↓ | TSPTW-50 Hard Infsb%↓ | Gap↓ | TSPTW-100 Medium Infsb%↓ | Gap↓ |
|---|---|---|---|---|---|---|
| Construction-only | 3.77% | 5.230% | 37.27% | 1.635% | 0.12% | 10.931% |
| w/o Shared Rep. | 0.01% | **2.047%** | 0.68% | 0.199% | 0.00% | 7.589% |
| w. Shared Rep. | **0.00%** | 2.173% | **0.01%** | **0.014%** | **0.00%** | **5.815%** |

Table 18: Model parameters and performance of different unifications on TSPTW-50 Hard. ✓ indicates shared representation.

| Encoder | Decoder | #Params | Infsb%↓ | Gap↓ |
|---------|---------|---------|---------|------|
| × | × | 2.8M | 0.68% | 0.199% |
| ✓ | ✓ | 1.4M | **0.01%** | 0.041% |
| ✓ | × | 1.6M | **0.01%** | **0.014%** |

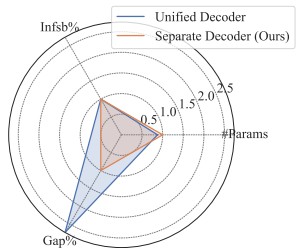

Figure 8: Effects of separate decoder.

We further evaluate this design on CVRPBLTW-100, where using separate encoders reduces the optimality gap slightly but increases model parameters by 1.7× compared to the unified encoder version of CaR (2.9M vs. 1.7M). To balance performance and computational efficiency, we advocate for the unified encoder design.

Moreover, we explore the unification of the encoder and decoder components. As shown in Table 18 and Figure 8, results on TSPTW-50 indicate that employing a separate decoder increases the number of model parameters by 1.1×, yet it further reduces the optimality gap. Therefore, this paper adopts an architecture with a unified encoder and separate decoders for each module.

We also examine the effect significance of shared representation compared to another learning component, i.e., the supervised loss. Results show that shared representation is mostly critical. Across all settings (regardless of the construction used, which is different from what we observe in the impact of the supervised loss), the shared representation plays the dominant role. It enables the necessary information synergy, allowing the refinement module to leverage structural context from the construction phase to repair constraints effectively.

Table 19: Comparison of different backbones with shared encoder and supervised loss.

| Backbones | Shared Rep. | $\mathcal{L}_{\text{SL}}$ | Gap ↓ | Infsb%↓ |
|-----------|-------------|---------------------------|-------|---------|
| **POMO* (Construction-only)** | – | – | 1.959% | 38.22% |
| CaR-POMO* | × | ✓ | 0.136% | 0.29% |
| CaR-POMO* | ✓ | × | 0.199% | 0.68% |
| CaR-POMO* (Ours) | ✓ | ✓ | 0.014% | 0.01% |
| **PIP (Construction-only)** | – | – | 0.177% | 2.67% |
| CaR-PIP | × | × | 0.156% | 1.85% |
| CaR-PIP | × | ✓ | 0.175% | 2.58% |
| CaR-PIP | ✓ | × | 0.006% | 0.01% |
| CaR-PIP (Ours) | ✓ | ✓ | **0.005%** | **0.00%** |

### E.5 DISCUSSION OF THE SHORT-HORIZON DESIGN

To further validate CaR's short-horizon design, we further retrain NeuOpt* by gradually decreasing the training horizon of NeuOpt and evaluate its performance under two inference budgets: (1) $T_test = 20$ (same as CaR), and (2) $T_test = 1,000$, which reflects the refinement length typically required by neural improvement solvers. Results in Table 20 show that NeuOpt* cannot produce any feasible solutions under a limited $T_test$, regardless of whether its training horizon is short or long, while CaR achieves strong performance, with an optimality gap of 0.005% and 0% infeasibility within just 20 steps of refinement. This discrepancy arises because NeuOpt* is inherently unsuitable for short-horizon refinement. Even when retrained with our designed constraint-related features and constraint-aware penalty function in the MDP formulation, NeuOpt* relies on long rollouts to gradually correct constraint violations and improve performance. In contrast, CaR is specifically designed to enable rapid feasibility refinement and solution improvement within tight step budgets. This highlights that short-horizon construct-and-refine is nontrivial and architecturally distinct from standard improvement-only methods.

Table 20: Performance of NeuOpt* with different short-horizon settings during training.

| Method | $T_{\text{train}}$ | $T_{\text{test}}$ | Gap | Infsb% |
|--------|--------------------|--------------------|------|--------|
| NeuOpt* | 5 (same as CaR) | 20 | / | 100% |
| | 10 | 20 | / | 100% |
| | 20 | 20 | / | 100% |
| | 100 | 20 | / | 100% |
| | 200 (original NeuOpt) | 20 | / | 100% |
| NeuOpt* | 5 (same as CaR) | 100 | / | 100% |
| | 10 | 100 | 1.314% | 99.79% |
| | 20 | 100 | 0.397% | 7.36% |
| | 100 | 100 | 0.145% | 0.22% |
| | 200 (original NeuOpt) | 100 | 0.061% | 0.19% |
| **CaR** | **5** | **20** | **0.005%** | **0.00%** |

## E.6 ADDITIONAL RESULTS FOR THE PATTERN OF CaR'S REFINEMENT

During inference, we follow (Kwon et al., 2020) to augment each instance 8×, generating eight initial solutions for refinement. All eight trajectories on a TSPTW instance are presented in Figure 9 to illustrate the diversity in initial solutions and their subsequent refinement. Moreover, to better understand how CaR refines solutions for feasibility and optimality, we also visualize the objective trajectories on other randomly generated TSPTW-50 instances in Figure 10. CaR dynamically navigates both feasible and infeasible regions, adjusting its focus based on the current solution state. For example, in the instance of Figure 10 (d), CaR begins with a near-feasible solution ($t = 0$), achieves feasibility in one step ($t = 1$), explores infeasible regions ($t = 2 - 12$), and ultimately improves optimality ($t = 13$). In contrast, in the instance of Figure 10 (c), the construction already yields a feasible, high-quality solution, and refinement continues to enhance optimality efficiently. These patterns illustrate CaR's ability to balance feasibility and optimality by exploring both feasible and infeasible spaces. By comparison, NeuOpt fails to escape infeasible regions within limited steps, highlighting CaR's advantage in constraint-aware refinement.

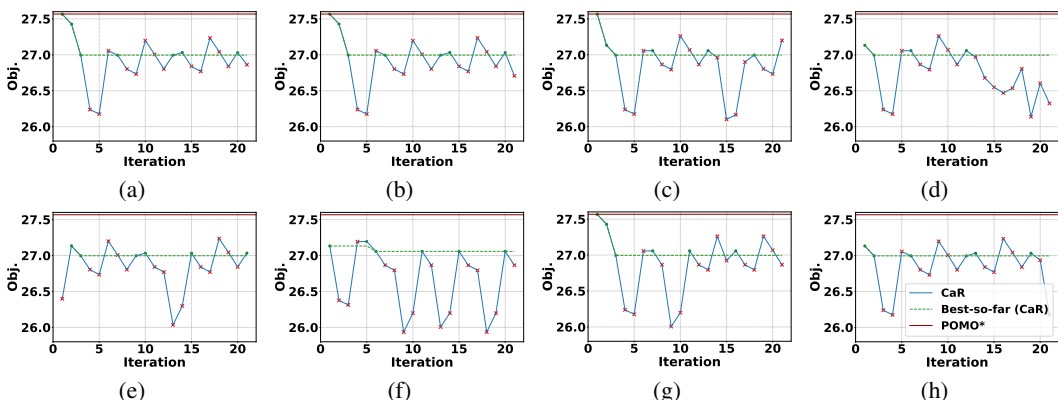

Figure 9: CaR's refinement on one TSPTW instance. Green dots and red crosses denote the feasible and infeasible solutions, respectively. Green dashed lines mark the best-so-far feasible objective; the red line indicates the best objective with data augmentation. CaR's refinement process is shown over time, with $t = 0$ representing the initially constructed solution.

## E.7 EFFECTS OF DIFFERENT NUMBERS OF INITIAL SOLUTIONS

Our CaR framework first samples $S$ solutions per instance during construction, then selects $p$ solutions for refinement over $T_R$ steps. As shown in Figure 11, reducing $p$ from 10 to 5 on TSPTW-50 increases both infeasibility and optimality gap. In contrast, further increasing it from 10 to 20 slightly enhance the performance (the optimality gap reducing from 0.014% to 0.007%, and the infeasibility reducing from 0.01% to 0.00%). This finding indicates that larger $p$ yields better final performance, but the gains become increasingly marginal as $p$ grows. Since $p$ is constrained only by GPU memory (we

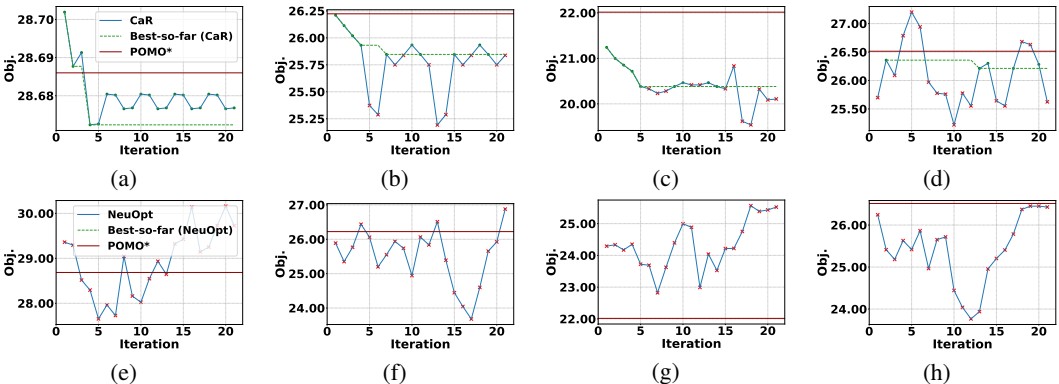

Figure 10: Refinement efficiency of CaR (*top*) and NeuOpt (*bottom*) on typical TSPTW-50 instances.

use a single Nvidia RTX 4090), we select the largest feasible $p$ in practice (i.e., $p=10$). Nonetheless, even with a smaller $p$, CaR consistently outperforms the state-of-the-art neural solver PIP (Bi et al., 2024), as well as classic heuristics, including LKH-3 with limited trials (100), ORTools, and greedy heuristics.

### E.8 HYPERPARAMTER EXPERIMENTS

We initially set hyperparameters without extensive tuning, focusing on validating the framework design. To assess their effects, we ran CaR-POMO* on TSPTW-50. (1) For diversity loss ($\alpha_1$), set at 0.01 to avoid dominating gradients, removing it degrades performance while larger values (e.g., 0.1) also harm learning. (2) For supervised loss ($\alpha_2$), set at 1, larger weights (e.g., 2) reduce performance by overemphasizing supervision. Overall, results show that CaR is robust to hyperparameters, and improvements reflect the framework's inherent strength rather than aggressive tuning.

Table 21: Hyperparameter sensitivity.

| Hyperparameters | Gap ↓ | Infsb% ↓ |
|---|---|---|
| $\alpha_1 = 0$ | 0.421% | 0.58% |
| $\alpha_1 = 0.01$ (Ours) | 0.014% | 0.01% |
| $\alpha_1 = 0.1$ | 1.483% | 32.68% |
| $\alpha_2 = 0$ | 0.136% | 0.29% |
| $\alpha_2 = 1$ (Ours) | 0.014% | 0.01% |
| $\alpha_2 = 2$ | 1.63% | 32.12% |

### E.9 SENSITIVITY TO RANDOM SEEDS AND STATISTICAL SIGNIFICANCE

**Sensitivity to random seeds.** To assess the robustness of CaR, we evaluate its performance under five different random seeds, i.e., 2023, 2024, 2025, 2026 and 2027, during inference. As shown in Table 22, the standard deviations are small, indicating that CaR's performance is stable with respect to random initialization. Note that we use 2023 as the default random seed for inference across all baselines to ensure consistency in performance comparison, following (Zhou et al., 2024).

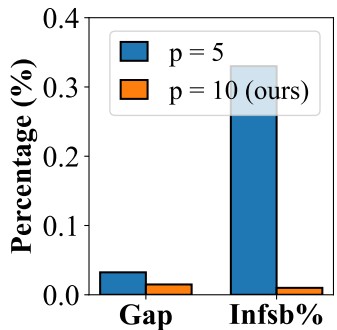

Figure 11: Effects of $p$.

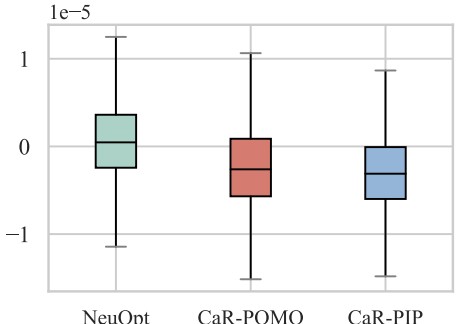

Figure 12: Boxplot of diverse methods on TSPTW-50.

Table 22: Standard deviation of the results in Table 2 under five different random seeds.

| Problem | Method | n=50 | | n=100 | |
|---|---|---|---|---|---|
| | | Gap ↓ | Infsb% ↓ | Gap ↓ | Infsb% ↓ |
| **TSPTW** | CaR-POMO ($T_R = 20$) | 0.016% ± 0.000% | 0.01% ± 0.00% | 0.401% ± 0.002% | 2.33% ± 0.02% |
| | CaR-PIP ($T_R = 20$) | 0.005% ± 0.000% | 0.00% ± 0.01% | 0.158% ± 0.001% | 0.64% ± 0.06% |
| **CVRPBLTW** | CaR ($k$-opt) ($T_R = 20$) | 2.166% ± 0.025% | 0.00% ± 0.00% | -1.697% ± 0.006% | 0.00% ± 0.00% |
| | CaR (R&R) ($T_R = 20$) | 0.454% ± 0.012% | 0.00% ± 0.00% | -2.473% ± 0.015% | 0.00% ± 0.00% |

**Statistical significance.** Figure 12 illustrates the distribution of optimality gaps for NeuOpt, CaR-POMO, and CaR-PIP, corresponding to the results on TSPTW-50 in Table 2. We additionally conduct significance tests, revealing that all pairwise differences are statistically significant ($p < 0.001$). This suggest that CaR delivers a significant performance improvement over the representative baselines.

### E.10 DISCUSSION ON THEORETICAL ANALYSIS

While our primary contribution is empirical, we acknowledge the importance of establishing strong theoretical foundations for NCO. In this section, we outline promising directions for analyzing the learnability, convergence, and generalization bounds of the proposed CaR framework. 1) **Learnability:** Since CaR jointly trains a construction module (policy $\pi_C$) and a refinement module (policy $\pi_R$) with a shared encoder, a key theoretical question concerns the sample complexity or PAC-style learnability of this joint policy class. Specifically, future work could investigate how many training instances are required for the learned $\pi_C$ and $\pi_R$ to yield solutions within $\epsilon$ of the optimal cost with a constraint-violation probability $\leq \delta$ (with high probability). Such analyses could leverage established results on policy gradient sample complexity Yuan et al. (2022). 2) **Convergence:** CaR employs entropy-regularized policy gradients (to encourage diversity) within a short-horizon refinement process. Consequently, convergence analysis could draw upon existing literature regarding smooth policy gradient methods Agarwal et al. (2021) and two-stage or bilevel RL optimization Thoma et al. (2024). These frameworks may help characterize the conditions under which joint updates to the shared encoder converge to a stationary point. 3) **Generalization Bounds:** The generalization behavior of CaR can be examined from multiple perspectives. One direction follows recent NCO generalization analyses, studying how a learned policy transfers across diverse settings (e.g., varying instance sizes, distributions, or constraint patterns). Alternatively, from the perspective of RL generalization theory, one could derive Rademacher-complexity-style bounds Duan et al. (2021) to formally characterize the gap between empirical and expected performance.

## F THE USE OF LARGE LANGUAGE MODELS (LLMs)

In accordance with ICLR policy, we disclose that LLMs were used solely to polish the writing and improve readability. LLMs were not involved in research ideation, experimental design, or any part of the scientific contributions of this work.

