# OpenReview forum: "Towards Efficient Constraint Handling in Neural Solvers for Routing Problems"
_ICLR.cc/2026/Conference — ICLR 2026 Poster_

### Official Review · Reviewer_FEWD · 2025-10-30

**Soundness:** 3
**Presentation:** 3
**Contribution:** 2
**Rating:** 6
**Confidence:** 2

**Summary:**

This paper proposes Construct-and-Refine (CaR), a framework designed to improve constraint satisfaction in neural combinatorial optimization solvers. Existing constraint-handling approaches can be broadly categorized into feasibility masking and feasibility awareness, but both tend to struggle with hard-to-satisfy constraints.
The proposed framework introduces two decoders: one for constructing an initial feasible solution and another for refining it iteratively. The authors evaluate CaR on TSPTW and CVRPLTW, demonstrating that it achieves higher constraint satisfaction rates for difficult instances and better objective values for larger-scale problems compared to existing methods.

**Strengths:**

- The proposed framework achieves superior constraint satisfaction and objective performance across multiple constraint types, demonstrating that the construct-and-refine approach can lead to more effective constraint handling.

- The paper includes a well-designed ablation study that helps clarify which components of the framework contribute most to performance improvements.

**Weaknesses:**

- The range of constraints for which the proposed framework is effective remains unclear. For example, constraints such as time-window or route-length can often be relaxed smoothly, but it is not evident whether CaR is equally effective for non-continuous or discrete constraints, such as precedence or ordering constraints.

**Questions:**

- What types of constraints are better handled by existing constraint-handling methods, and which types particularly benefit from the proposed CaR framework?

- What characteristics must existing methods, such as POMO, have in order to integrate with CaR?

- In what situations does the proposed method fail to find a constraint-satisfying solution? Are there any common patterns or characteristics in such cases?

---

> ### Author Response · Authors · 2025-11-24
> **Response to Reviewer FEWD (1/3)**
>
> We truly appreciate your constructive and insightful comments. We are delighted that the reviewer recognizes our effectiveness in constraint handling and well-designed ablation studies. We understand your main concern lies in the applicability range of CaR and hope the response below could successfully address it.
>
> ---
>
> **\[R1: When CaR is effective (W1, Q1)\]** Thank you for the insightful question\!
>
> (1) We first show whether CaR is equally effective for non-continuous or discrete constraints. To verify if CaR is effective for discrete constraints, **we implemented CaR on the Sequential Ordering Problem (SOP)** \[1\], a routing problem defined by precedence constraints.
>
> * **Problem Definition:** defined on a graph $\\mathcal{G} \= \\{\\mathcal{V}, \\mathcal{E}\\}$. The objective is to find a permutation $\\tau= \\{ v\_1, …, v\_n\\}$ with fixed start and end nodes $v\_1$ and $v\_n$ that minimizes the total travel cost, subject to all precedence constraints $(v\_i, v\_j) \\in \\mathcal{P}$, i.e., $v\_i$ must precede $v\_j$.
> * **Instance Generation:** To ensure feasibility, we sample a random valid permutation, identifying all implied precedence pairs. We then randomly sample h% of these pairs as constraints using a mixture of pairwise Euclidean distance (weight g) and random noise (weight 1−g). We evaluate CaR on two SOP variants: Variant 1 (h \= 20, g \= 0.3) and Variant 2 (h \= 20, g \= 0.8).
> * **Feasibility Masking:** For SOP, masking during construction is tractable (masking nodes whose predecessors are unvisited).
> * **Results:** Across both variants, CaR consistently outperforms strong neural baselines, achieving **53-82% relative improvement in solution quality** over POMO.
>
> **Table R7: Results on SOP-50 with discrete precedence constraints.**
>
> | Method | SOP Variant 1 |  | SOP Variant 2 |  |
> | :---: | :---: | :---: | :---: | :---: |
> |  | Obj | Gap | Obj | Gap |
> | LKH-3 | 14.732 | \* | 16.302 | \* |
> | POMO | 14.943 | 1.436% | 16.376 | 0.463% |
> | CaR | 14.831 | **0.676%** | 16.316 | **0.084%** |
> | | | | | |
>
> This confirms that **CaR can still be effective on non-continuous or discrete constraints**, which aligns well with our goal to design a general-purpose framework for constrained VRPs.
>
> (2) We then discuss the range of the constraints where our CaR is effective. While the precise effectiveness boundary requires empirical validation, we offer the following practical insight from our experience: **CaR is applicable across most types of constraints, whether continuous or discrete, but is particularly effective when the constraints are hard.** Empirically, CaR demonstrates more significant improvements in both feasibility and solution quality when the constraints are hard. For example, as shown in Table R8 below (derived from the original Table 13), CaR’s improvement is more significant in harder constraint TSPTW settings. This suggests CaR is most effective when construction alone fails; in contrast, CaR leverages learning-based feasibility refinement and shared representations to enhance joint decision-making, efficiently addressing the structural flaws that inherently limit one-shot generation in prior methods.
>
> **Table R8: Results on TSPTW-50 with different constraint hardness.**
>
> | Method | TSPTW-50 (Easy) |  | TSPTW-50 (Medium) |  | TSPTW-50 (Hard) |  |
> | :---: | :---: | :---: | :---: | :---: | :---: | :---: |
> |  | Gap | Infsb% | Gap | Infsb% | Gap | Infsb% |
> | POMO\* |  3.08% | 0.00% | 5.23% | 3.77% |  1.635% | 37.27% |
> | CaR | **2.59%** | **0.00%** | **2.17%** | **0.00%** | **0.014%** | **0.01%** |
> |   |   | |  | |   |   |
>
> (3) To answer “what types of constraints are better handled by existing constraint-handling methods”, we categorized VRPs into three types based on constraint hardness:
>
> 1. Simple VRPs, where feasibility masking is tractable and effective (e.g., CVRP),
> 2. Complex VRPs, where masking is tractable but overly restrictive, thus ineffective (e.g., CVRPBLTW),
> 3. Complex VRPs, where computing feasibility masks is NP-hard or intractable (e.g., TSPTW, TSPDL).
>
> As shown in the original Table 11, the existing SoTA solver SGBS already achieves an optimality gap of 0.170% on CVRP-100. While CaR further reduces it to 0.148%, the improvement is marginal compared to more complex VRPs like CVRPBLTW, where CaR significantly reduces the optimality gap from SGBS’s 0.854% to \-2.448%. This highlights that when constraints are simple, they can be effectively handled by existing methods like masking. A more detailed analysis is provided below in `Response (2/3)`.

---

> ### Author Response · Authors · 2025-11-24
> **Response to Reviewer FEWD (2/3)**
>
> (Continued)
>
> | Constraint-handling Method |  | Most Effective For | Limitations |
> | :---- | :---- | :---- | :---- |
> | **Feasibility Masking** *(existing)* |  | Simple constraints (e.g., CVRP), validated in Table 8: removing masking increases gap from 0.857% to 0.921% | 1\. NP-hard to compute on problems like TSPTW 2\. Overly restrictive in CVRPBLTW (e.g., masks \>60% of nodes) |
> | **Feasibility awareness** | Feature *(existing)* | Either simple or complex constraints (essentials for learning-based methods) | Only auxiliary; lacks corrective power |
> |  | Reward/Penalty *(existing)* | Simple constraints | The guidance of reward/penalty loses effectiveness in complex VRPs, as verified in \[2\]. |
> |  | Shared representation *(**CaR**)* | Complex  constraints, by promoting cross-module constraint awareness | Moderate gains in simple VRPs where constraints are easily handled |
> | **Feasibility refinement** *(**CaR**)* |  | Complex constraints, where both feasibility and optimality must be addressed jointly | Less impactful when feasibility is trivial and optimality is the only target |
> | | | | | | |
>
> ```
> Reference:
> [1] A heuristic manipulation technique for the sequential ordering problem. Computers & Operations Research, 2008.
> [2] Learning to handle complex constraints for vehicle routing problems. NeurIPS, 2024.
> ```
>
> ---
>
> **\[R2:  What must existing methods have to integrate with CaR? (Q2)\]** We thank the reviewer for the insightful question. CaR is designed as a **modular construct-and-refine framework**, which means it can accommodate a wide range of existing methods as long as they meet basic functional roles: 1\) a construction module to generate diverse and high-quality initial solutions, and 2\) a refinement module to improve such solutions. **In principle, most existing construction or improvement heuristics can be integrated into CaR with some modifications**. For example, we evaluated it by pairing POMO and PIP as construction backbones with NeuOpt and N2S as refinement modules. Moreover, to fully leverage CaR’s *shared representation* design, it is beneficial for the construction and improvement module to **share similar neural architectures**, especially in the encoder or early layers. We highlight that it is easy to meet since the encoders of both modules aim to represent the problem structure itself. For example, in scheduling, existing neural construction \[3\] and improvement \[4\] heuristics both adopt GNN-based architectures. We believe this represents a promising direction for future work and will include this discussion in the revised manuscript.
>
> ```
> References:
> [3] Learning to dispatch for job shop scheduling via deep reinforcement learning. NeurIPS, 2020.
> [4] Deep Reinforcement Learning Guided Improvement Heuristic for Job Shop Scheduling. ICLR, 2024.
> ```

---

> ### Author Response · Authors · 2025-11-24
> **Response to Reviewer FEWD (3/3)**
>
> **\[R3: Patterns when CaR fails (Q3)\]** Thank you for the interesting and constructive comment\!
>
> First, we note that **CaR achieves near-perfect feasibility** across all benchmarks. For example, it attains 0% infeasibility on TSPTW-50, CVRPBLTW-50, and CVRPBLTW-100, and only 0.02% infeasibility on TSPTW-100. Nonetheless, we observe a slightly higher infeasibility rate (2.34%) when replacing the strong construction backbone (PIP) with **a weaker one (POMO)**. We thus use this setting as a case study to analyze failure patterns. To identify common patterns, we partitioned the test instances into two distinct groups based on the solver's outcome: **Feasible Subset:** Instances where CaR successfully found a feasible solution; and **Infeasible** **Subset:** Instances where CaR failed to find a valid solution (i.e., remained infeasible). We conduct statistical significance analysis (p-value \< 0.01) to identify differences in instance features and solution behaviors across the two sets.
>
> Our analysis reveals two key observations:
>
> 1. **Structural heterogeneity**: Infeasibility tends to occur in instances with more heterogeneity of node distribution and time window distribution.
>    * **Sparser node distributions**, reflected by larger average pairwise node distances and higher variance.
>    * **Dispersed time window distributions**, reflected by a lower overlap percentage of the time window between nodes.
> 2. **Solving patterns**: Instances with infeasible solutions often start with weaker constructions and demand longer refinement.
>    * **Higher constraint violation at initialization**: Instances with infeasible solutions begin with an average violation of 3.36, reduced to 2.21 after refinement, whereas feasible ones start at 1.76.
>    * **Higher refinement demand**: As shown in Table 2, CaR-POMO’s infeasibility rate decreases from 4.20% to 2.34% when increasing the refinement steps from 5 to 20\. Further extending to 80 steps only marginally lowers the infeasibility to 2.1%. However, increasing the construction sample size (4×) brings more significant improvement, reducing infeasibility to 1.52%. This aligns with our design philosophy that high-quality construction is more critical than prolonged refinement under a lightweight improvement budget
>
> In summary, **CaR’s failures in generating feasible solutions typically arise in instances with more heterogeneous node and time window distributions, particularly when the initial construction is weak, and the refinement horizon is insufficient.** We highlight that when we switch the backbone from the weak POMO\* to the strong PIP and run 20 refinement steps, the infeasibility almost disappears (as shown in the new Table 7 in the revised manuscript).
>
> We are inspired by these findings\! A promising future direction is to develop adaptive strategies, such as instance-aware dynamic refinement lengths or constraint-weighted losses, that automatically adjust effort based on the detected hardness of the instance. This would further enhance the performance of CaR. Thank you again for the insightful suggestion\!
>
> ---
>
> We sincerely appreciate the reviewer’s thoughtful evaluation and constructive feedback. If there are still aspects of our explanation that appear ambiguous or incomplete, we would be glad to clarify them in more detail. We welcome any further questions or suggestions and are fully committed to improving the manuscript wherever needed.

---

> > ### Comment · Reviewer_FEWD · 2025-11-26
> >
> > Dear Authors,
> >
> > Thank you for the additional experiments and the careful explanations addressing my concerns. I feel that my understanding of the characteristics of the problems for which CaR is effective, the types of constraints it can handle, and its compatibility with existing methods has deepened considerably. As most of my main concerns have been resolved, I intend to maintain my positive evaluation.

---

> ### Author Response · Authors · 2025-11-26
> **Thank you for your continued support!**
>
> Dear Reviewer FEWD,
>
> We are delighted that our additional experiments and explanations have successfully resolved your main concerns.
>
> We truly appreciate your positive feedback and your continued support!!!
>
> Best regards,
> Authors

---

### Official Review · Reviewer_8D1i · 2025-10-30

**Soundness:** 3
**Presentation:** 2
**Contribution:** 3
**Rating:** 8
**Confidence:** 4

**Summary:**

This paper proposes a neural combinatorial optimization (NCO) solver for vehicle routing problems (VRPs) with hard constraints.
Existing NCO approaches can be categorized into (1) construction solvers that sequentially build solutions, (2) improvement solvers that iteratively refine them, and (3) hybrid methods that combine both.
However, these approaches struggle to effectively handle hard constraints such as time windows or multiple interdependent resource limits.
To address this issue, the authors introduce a new framework called Construct-and-Refine (CaR), which explicitly refine a constructed solution to recover feasibility.
CaR jointly trains construction and refinement modules with a shared encoder, enabling fast and effective recovery of feasibility while maintaining solution quality.
Experiments on several constrained VRP variants (TSPTW, CVRPBLTW) demonstrate that CaR achieves superior feasibility, optimality, and computational efficiency compared to both classical heuristics and recent neural baselines.

**Strengths:**

- This paper is well-motivated. It addresses a fundamental challenge in neural combinatorial optimization (NCO) — effectively satisfying hard constraints such as time windows constraints. This limitation has been a core weakness of existing NCO solvers, and this paper clearly explains the importance of effectively handling such constraints.
- The proposed approach is conceptually sound. Introducing an explicit feasibility refinement module to recover infeasible solutions is a natural and intuitive.
- Though the proposed approach is simple, it works well in practice. Experiments demonstrate that it consistently finds feasible solutions more reliably than other NCO solvers and classical heuristics on hard-constrained routing problems such as TSPTW and CVRPBLTW, while also achieving higher solution quality. The experimental evaluation is comprehensive, covering multiple benchmarks, ablation studies, and generalization tests that convincingly support the method’s effectiveness.

**Weaknesses:**

While the paper’s empirical contributions are strong, there is no theoretical discussion of why the joint learning of construction and refinement works well. In particular, questions regarding learnability, convergence, or generalization bound of the proposed method remain open. Formal analysis would strengthen the work’s long-term impact.

**Questions:**

- The first question is how CaR could be analyzed from a theoretical standpoint (possibly as a future work). Possible directions include learnability, convergence, or generalization bound.
- The second question is about applicability of the proposed method. It appears general and could be extended to other structured optimization domains where constraint handling is non-trivial, such as scheduling problems or SAT.

---

> ### Author Response · Authors · 2025-11-24
> **Response to Reviewer 8D1i (1/2)**
>
> We sincerely thank the reviewer for the insightful feedback and the strong support of our work. We are pleased that the work is recognized as well-motivated, conceptually sound, and empirically validated through comprehensive experiments. We understand that the remaining concern pertains to the theoretical discussion and general applicability of CaR beyond the current settings. We hope the following response will effectively address this point.
>
> ---
>
> **\[R1: Theoretical discussion (W1, Q1)\]** We sincerely appreciate this feedback. We fully agree that a formal theoretical analysis would significantly enhance the rigor and long-term value of this line of research. While a comprehensive theoretical derivation is outside the immediate scope of this empirical study, we see it as an exciting and necessary direction for future work. Below, we provide some possible theoretical perspectives on the learnability, convergence, or generalization bound analysis of our proposed CaR:
>
> - **Learnability:** In CaR, we propose to jointly train a construction module (policy $\\pi_C$) and a refinement module (policy $\\pi_R$) with a shared encoder. A promising direction for future work is to investigate the sample-complexity or PAC-style learnability of the joint construction–refinement policy class, e.g., *how many training instances are required for the learned $\\pi_C$, $\\pi_R$ to, with high probability, yield solutions within $\\epsilon$ of optimal cost and with constraint‐violation probability ≤ $\\delta$.* Such analyses could potentially leverage established results on policy gradient sample complexity [1].
> - **Convergence:** Since CaR uses entropy-regularized policy gradients (diversity loss) and a short-horizon refinement process, convergence analysis could draw on results from smooth policy gradient methods \[2\] and from two-stage or bilevel RL optimization \[3\]. This may help characterise when joint updates to the shared encoder converge to a stationary point.
> - **Generalization bound:** The generalization behavior of CaR can be examined from multiple angles. One direction follows the line of NCO generalization analyses, which study under what conditions and to what extent a learned policy generalizes across diverse settings (e.g., instance sizes, distributions, or constraint patterns). Another perspective is based on RL generalization theory, where one could derive Rademacher-complexity-style bounds \[4\] to characterize the gap between empirical and expected performance.
>
>
> Finally, we acknowledge that the field of neural combinatorial optimization (NCO) has largely been driven by empirical advances aimed at narrowing the gap between neural and classical solvers, yet it still lacks strong theoretical foundations. As recognized by all reviewers, our work demonstrates clear empirical superiority, surpassing both neural and classical baselines on hard-constrained VRPs, which may lay a solid basis for future theoretical exploration.
>
> ```
> References:
> [1] A general sample complexity analysis of vanilla policy gradient. AISTATS, 2022.
> [2] On the Theory of Policy Gradient Methods: Optimality, Approximation, and Convergence. JMLR, 2021.
> [3] Contextual Bilevel Reinforcement Learning for Incentive Alignment. NeurIPS, 2024.
> [4] Risk Bounds and Rademacher Complexity in Batch Reinforcement Learning. ICML, 2021.
> ```

---

> ### Author Response · Authors · 2025-11-24
> **Response to Reviewer 8D1i (2/2)**
>
> **\[R2: Applicability of CaR to other optimization domains (Q2)\]** Thank you for recognizing the generality of CaR. We believe that CaR is well-suited for broader structured optimization domains, particularly those where feasibility handling is challenging. To further verify if CaR is effective for other constraints, **we implemented CaR on the Sequential Ordering Problem (SOP)** \[1\], a discrete precedence-constrained routing task closely related to scheduling.
>
> * **Problem Definition:** defined on a graph $\\mathcal{G} \= \\{\\mathcal{V}, \\mathcal{E}\\}$. The objective is to find a permutation $\\tau= \\{ v\_1, …, v\_n\\}$ with fixed start and end nodes $v\_1$ and $v\_n$ that minimizes the total travel cost, subject to all precedence constraints $(v\_i, v\_j) \\in \\mathcal{P}$, i.e., $v\_i$ must precede $v\_j$.
> * **Instance Generation:** To ensure feasibility, we sample a random valid permutation, identifying all implied precedence pairs. We then randomly sample h% of these pairs as constraints using a mixture of pairwise Euclidean distance (weight g) and random noise (weight 1−g). We evaluate CaR on two SOP variants: Variant 1 (h \= 20, g \= 0.3) and Variant 2 (h \= 20, g \= 0.8).
> * **Feasibility Masking:** For SOP, masking during construction is tractable (masking nodes whose predecessors are unvisited).
> * **Results:** Across both variants, CaR consistently outperforms strong neural baselines, achieving **53-82% relative improvement in solution quality** over POMO.
>
> **Table R6: Results of SOP with discrete precedence constraint**
> | Method | SOP Variant 1 |  | SOP Variant 2 |  |
> | :---: | :---: | :---: | :---: | :---: |
> |  | Obj | Gap | Obj | Gap |
> | LKH-3 | 14.732 | \* | 16.302 | \* |
> | POMO | 14.943 | 1.436% | 16.376 | 0.463% |
> | CaR | 14.831 | **0.676%** | 16.316 | **0.084%** |
> |  | |  |  | |
>
> In this paper, we study routing problems due to the availability of mature open-source baselines and datasets. Since CaR is designed as a **modular construct-and-refine framework**, which means it can accommodate a wide range of existing methods as long as they meet basic functional roles: 1\) a construction module to generate diverse and high-quality initial solutions, and 2\) a refinement module to improve such solutions. This is inherently general, as the paradigm of initial solution generation followed by iterative search is fundamental to most heuristic approaches in combinatorial optimization. Moreover, to fully leverage CaR’s *shared representation* design, it is beneficial for the construction and improvement module to share similar neural architectures, especially in the encoder or early layers. We highlight that it is easy to meet since the encoders of both modules aim to represent the problem structure itself. For example, in scheduling, existing neural construction \[5\] and improvement \[6\] heuristics both adopt GNN-based architectures. This shared architectural foundation makes it straightforward to integrate CaR’s design for more efficient constraint handling for hard-constrained instances. We believe this represents a promising direction for future work and will include this discussion in the revised manuscript. Thank you again for the valuable feedback\!
>
> ```
> References:
> [5] Learning to dispatch for job shop scheduling via deep reinforcement learning. NeurIPS, 2020.
> [6] Deep Reinforcement Learning Guided Improvement Heuristic for Job Shop Scheduling. ICLR, 2024.
> ```

---

> > ### Comment · Reviewer_8D1i · 2025-11-28
> > **Thank you for the detailed responses**
> >
> > Thank you very much for providing such thorough and thoughtful answers to my questions. Your clarifications have resolved most of my concerns. I believe that incorporating the responses to these two questions into the appendix would be valuable for future research, and I would kindly encourage you to include them in the revised version.

---

> ### Author Response · Authors · 2025-11-28
> **Thank you so much for your strong support!**
>
> Thank you so much for your strong support! We deeply appreciate your recognition of our rebuttal. We are more than happy to include these insights in our revised version. We have included the outline of the answer to the first question in the Future Work (Section 6), and provided the details in Appendix E.10 as you suggested. Regarding the answer to the second question, we have incorporated it into Section 5.1 and Appendix B.5.
>
> Thank you again for your positive feedback.

---

### Official Review · Reviewer_iA9w · 2025-11-01

**Soundness:** 2
**Presentation:** 2
**Contribution:** 2
**Rating:** 4
**Confidence:** 4

**Summary:**

This paper addresses the challenge of applying neural combinatorial optimization (NCO) solvers to VRPs with complex constraints. The authors introduce "Construct-and-Refine" (CaR), a hybrid framework that jointly trains a construction and refinement module. The construction module is trained to produce diverse and high-quality solutions. These are then fed to a refinement module that operates for a short horizon to rapidly find feasible, high-quality solutions. Experiments on hard-constrained VRPs (TSPTW, CVRPBLTW) and others (CVRP, TSPDL) show that CaR can achieve state-of-the-art feasibility and solution quality.

**Strengths:**

(1) The paper is well-written and clearly structured.

(2) The paper tackles a critical problem. The failure of NCO solvers to handle complex constraints is arguably the barrier to their practical adoption.

(3) The experimental evaluation is comprehensive, demonstrating strong results on different VRPs with hard constraints.

**Weaknesses:**

(1) This paper proposes CaR to efficiently handle complex constraints in VRPs and subsequently produce high-quality solutions. However, the core contribution appears to combine PIP and existing hybrid methods (LCP and NCS) for solving hard-constrained VRPs.

(2) The idea of combining construction and improvement is not new. NCS also employs a joint training paradigm with a shared component. The distinction of the joint training paradigm is not strong.

(3) What is the fundamental difference between the approach used in this paper for handling constraints and the essence of PIP?

(4) The ablation study of the effectiveness of the joint training is unconvincing. A much stronger ablation would be jointly training PIP + NeuOpt, but with separate encoders and without the $L_{SL}$ feedback loss, and compare with (1) add the $L_{SL}$ loss, (2) add the  shared encoder

(5) The arguments in this paper "In CVRPBLTW, for instance, strict masking filters out more than 60% of nodes, severely limiting the search space (Figure 6) and hindering RL convergence." Why does limiting the model search space hinder RL convergence?

**Questions:**

(1) Can the authors more clearly distinguish the conceptual difference between CaR's joint training and that of NCS? Both seem to jointly train a constructor and improver with a shared component.

(2) Could the authors provide results for CaR compared against NeuOpt-GIRE (or another SOTA improvement solver)  that is re-trained to optimize its performance for a short rollout?

(3) Why was the ablation in Table 5 compared against a "Random" constructor? Could the authors provide results for CaR compared with a jointly trained constructor and improver with a separate encoder and the same model, but not use the $L_{SL}$ loss during training?

(4) How sensitive is the framework to the choice of p (top-p solutions)?

---

> ### Author Response · Authors · 2025-11-24
> **Response to Reviewer iA9w (1/4)**
>
> We sincerely appreciate the reviewer’s constructive feedback. We are pleased that the reviewer recognizes our work as well written, tackles a critical problem, and achieves comprehensive and strong results across different hard-constrained VRPs.  We hope the following response and the additional experiments will adequately address your remaining concerns.
>
> ---
> **\[R1: Combine PIP and LCP/NCS (W1)\]** We thank the reviewer for this comment. We believe this concern may stem from some misunderstandings regarding the capabilities and design principles of CaR versus LCP and NCS. We truly appreciate the opportunity to clarify it.
>
> While we agree with the reviewer that all these methods can be classified as "hybrid" methods (combining construction and search), we emphasize that such similarity does not imply equivalence in mechanism. **Fundamentally, these frameworks operate under opposing assumptions to achieve different goals.** Specifically, LCP and NCS aim to reduce optimality gaps under the assumption that feasibility is easily handled (via masking) and that very long runtimes are acceptable (e.g., tens of hours or days for 100-node CVRP/PDP). Consequently, their improvement modules are designed to be the primary drivers of optimality, lacking any specialized mechanisms for complex feasibility handling and solving efficiency. In contrast, CaR explicitly targets hard-constraint VRPs (TSPTW and CVRPBLTW) that current SoTA NCO solvers fail, and it prioritizes solving efficiently, which is designed to operate under a strictly limited runtime budget (less than a few minutes).
>
> Crucially, to validate that a trivial adaptation would not work well, we evaluated the direct combination of PIP with LCP/NCS and found that it results in **100% infeasibility** for these hard-constrained problems. Specifically, LCP breaks feasibility during decomposition due to its divide-and-conquer nature, while NCS fails to generate feasible solutions on hard-constrained TSPTW due to its original design focus on optimality under simple constraints. Thus, **both conceptually and empirically, CaR is a distinct solution to a distinct research problem.**
>
> We hereby restate our contributions. Our paper represents a few **“firsts”** for the NCO community that are not discussed or proposed in any prior work to the best of our knowledge:
>
> - **A new constraint handling scheme**: beyond *feasibility masking* and implicit *feasibility awareness*, CaR proposes *explicit feasibility refinement*, which is a novel perspective that seeks learning-based repairing of infeasible solutions in a few post-construction steps while preserving optimality (see Table 1 for a conceptual comparison). This overcomes the limitations of existing methods, which often fail when complex constraints make one-shot valid construction exponentially difficult.
> - **An efficiency-preserving joint-training framework:** unlike LCP/NCS, CaR proposes an efficiency-preserving joint training framework that constructs diverse, high-quality solutions well-suited for rapid refinement, guided by our redesigned MDP and the tailored loss functions. (see `R2` below for details).
> - **A novel feasibility-awareness mechanism via shared representation**: We introduce a cross-module learning mechanism via shared encoders that strengthens constraint handling beyond standard feature or penalty-based methods. By enabling joint gradient flow, the model learns simultaneously from both "creation" and "fix" perspectives, generating deep, constraint-aware representations essential for complex problems.
> - **SoTA performance:** On TSPTW-100, CaR reduces the gap from 1.026% to 0.146% and infeasibility from PIP’s 4.47% to 0.02%, even outperforming LKH-3 (0.13% infeasibility). On CVRPBLTW-100, CaR achieves a -2.448% gap vs. OR-Tools, outperforming NeuOpt (39-51% infeasibility) and SGBS (0.854% gap). See Figure 3 for detailed comparisons.

---

> ### Author Response · Authors · 2025-11-24
> **Response to Reviewer iA9w (2/4)**
>
> **\[R2: Conceptual Differences from NCS (W2, Q1)\]** Thank you for the question\! As we discussed above, while both NCS and CaR combine construction and improvement, their goals, assumptions, technical designs, and applicabilities are largely different. Below, we further analyze the key differences regarding their joint training frameworks and the shared component.
>
> **NCS: Infinite-horizon joint training for optimality**
>
> * **Goal & MDP:** NCS targets optimality improvement under the assumption of prolonged runtime. It formulates the problem as an infinite-horizon MDP trained with PPO.
> * **Shared component (Critic value):** Construction in NCS serves only to initialize solutions for long-term refinement. As both modules aim to reduce the optimality gap, NCS uses a shared critic for value estimation, but does not share policy network architectures (encoders/decoders) between modules and relies solely on simple feasibility masking, which fails under hard constraints.
>
> **CaR: short-horizon joint training for optimality and feasibility**
>
> * **Goal & MDP:** CaR targets efficiency-preserving constraint handling by formulating a short-horizon MDP that follows a “construct-then-fix” strategy in a few steps. To match this design, we remove the critic and apply REINFORCE, treating each refinement step equally. This allows us to optimize both optimality and feasibility by incorporating a penalty loss for both phases.
> * **Shared component (representation):** Unlike NCS, CaR uses a unified shared encoder, as hard constraints complicate embedding learning. This enables **information synergy**: constraint-aware representations learned during construction are reused in refinement. **It creates a deep "feasibility awareness" unavailable in NCS**, allowing the improvement module to **understand the structural reasons for a construction failure and repair it efficiently.**
>
> **Finally, we note that these shared components are specific to their tasks.** Introducing NCS’s shared critic into CaR’s short-horizon MDP may introduce computational waste to learning another critic network without benefit. Conversely, applying CaR’s shared representation to NCS would likely fail due to imbalance: with NCS’s long improvement horizon, gradients would be dominated by the improvement module, washing out construction features.
>
> All the above analyses reflect distinct design goals: **CaR is built for efficient and effective constraint handling under practical rollout limits, while NCS is not.**
>
> ---
> **\[R3: Differences from PIP (W3)\]** Thank you for the valuable comment. The fundamental differences are mainly fourfold as follows:
>
> * **Different roles of the construction module for constraint handling.** PIP relies entirely on the construction process for constraint handling: its auxiliary decoder must produce solutions that are both high-quality and feasible. This becomes impractical for hard-constraint VRPs where feasibility signals are sparse and step-wise feasibility satisfaction is highly restricted. In contrast, CaR decouples these goals: the constructor focuses on generating high-quality solutions (not necessarily feasible) that are deliberately structured to be easy for a lightweight refinement module to repair in a few targeted steps. This separation is crucial for both efficiency and feasibility.
> * **CaR proposes a new constraint handling scheme.** PIP is a supplement to feasibility masking for a specific subset of complex VRPs where masking is NP-hard. In contrast, CaR introduces a different constraint-handling scheme based on explicit feasibility refinement and synergised feasibility awareness via cross-paradigm representation learning between construction and refinement modules.
> * **Different applicability.** PIP is a specialized construction method designed for settings where feasibility masking is NP hard or intractable, such as TSPTW. When masking is tractable, as in multi-constraint CVRPBLTW or even standard CVRP, the designs in PIP essentially collapse to classical POMO and offer no additional benefit. Thus, PIP targets a narrow class of problems, whereas CaR is more general and can handle VRPs across a wide range of constraint hardness, including TSPTW, CVRP, CVRPBLTW, and more.
> * **CaR is compatible with PIP, but not vice versa.** PIP can be one of the backbones of CaR’s construction module. In this work, we validate CaR’s generality by using either PIP or POMO as construction backbones, and NeuOpt (k-opt) or N2S (R\&R) as improvement backbones, showing our modular and extensible design.
>
> **Finally, we note that CaR consistently outperforms PIP in both solution quality and feasibility.** For example, on the most challenging TSPTW-100 instance, CaR reduces the infeasibility rate from 4.47% to 0.02% and the optimality gap from 1.026% to 0.146% (see Tables 2 and 3 for more results).

---

> ### Author Response · Authors · 2025-11-24
> **Response to Reviewer iA9w (3/4)**
>
> **\[R4: Ablation of joint training (W4, Q3)\]** Thank you for the insightful suggestion\! To provide a complete picture, we first clarify the logic of our original ablations and then present the new results based on your specific request.
>
> We design the ablations in the following logic:
>
> * **Effects of construction/improvement combinations**: We assess alternative combinations of different construction (random or pretrained PIP) and improvement methods (untrainable LKH-3, pretrained NeuOpt with long horizons, and our trained NeuOpt with short horizons). **Table 5** shows that without joint training, these combinations are ineffective.
> * **Effects of joint training loss:** We ablate the diversity loss ($L\_{\\text{div}}$) and the supervised feedback loss ($L\_{\\text{SL}}$). **Table 4** shows that promoting diversity in the construction policy is crucial for overall performance, and supervised feedback further helps when the construction module is weak (see results in Table R3 below).
> * **Effects of shared representation: Table 6** demonstrates that shared representation is particularly beneficial when constraints are hard.
>
> We agree that jointly assessing the shared representation and supervised loss is a valuable addition. Following your suggestion, we added additional ablation of PIP+NeuOpt as follows and merged it with our original ablations of POMO\*+NeuOpt. Results indicate that:
>
> * **Shared representation is mostly critical**: Across all settings (regardless of the construction used), the shared representation plays the dominant role. It enables the necessary information synergy, allowing the refinement module to leverage structural context from the construction phase to repair constraints effectively.
> * **The impact of supervised Loss ($\\mathcal{L}\_\\text{SL}$ ) depends on the strength of the construction backbone:**
>     - With strong construction (PIP): adding $\\mathcal{L}\_\\text{SL}$  yields only marginal gains since PIP is already highly optimized (0.177\% gap, 2.67\% infeasibility), thus the feedback signal is less critical.
>     - With weak construction (POMO\*): standalone POMO* performs poorly (1.959% gap, **38.22% infeasibility**). In this scenario, joint training with $L\_{SL}$ dramatically improves performance to a 0.199% gap and **0.68% infeasibility**.
>
> **Table R3: Effects of the shared representation and supervised loss in CaR.**
>
> | Backbones | Shared Rep. / Unified encoder | Supervised loss ($\\mathcal{L}\_\\text{SL}$ ) | Gap | Infeasibility% |
> | :---: | :---: | :---: | :---: | :---: |
> | POMO\* (Construction-only) |  |  | 1.959% |  38.22% |
> | POMO+NeuOpt | x | √ | 0.199% | 0.68% |
> | POMO+NeuOpt | √ | x | 0.136% | 0.29% |
> | POMO+NeuOpt (CaR)| √ | √ | 0.014% | 0.01% |
> | PIP (Construction-only) |  |  | 0.177% | 2.67% |
> | PIP+NeuOpt | x | x | 0.156% | 1.85% |
> | PIP+NeuOpt | x | √ | 0.175% | 2.58% |
> | PIP+NeuOpt | √ | x | 0.006% | 0.01% |
> | PIP+NeuOpt (CaR)| √ | √ | **0.005%** | **0.00%** |
> |  |  |  |  |   |
>
> We have included the above new results and the discussion in Appendix E.4, highlighted in pink. We sincerely thank the reviewer for the suggestion.
>
> ---
> **\[R5: Why does strict masking in CVRPBLTW hinder RL convergence? (W5)\]** Thank you for the valuable questions\! We are sorry for the confusion, and we will make the statement more detailed. To clarify, we argue that **overly strict masking**, as observed in CVRPBLTW (Figure 6, where over 60% of candidate nodes are masked out during construction), **hinders exploration and convergence of RL toward a high-quality policy.**
>
> - To support this, we include additional results in **the new Figure 7** of the revised paper comparing two model variants on CVRPBLTW-100:
>   1. The strict-masked variant (red) converges rapidly in terms of training loss (saturating after \~2k epochs) but yields noticeably lower-quality solutions.
>   2. The relaxed-masking variant (blue) converges more gradually but ultimately attains significantly better solution quality.
>
> - These findings are consistent with prior literature [1,2], which shows that excessive action-space pruning leads to conservative policy behavior and degraded learning performance. We emphasize that feasibility masking remains effective and often essential for simpler VRPs (e.g., CVRP, see Appendix E.1), where constraints are less restrictive and the masked space still allows for sufficient exploration. However, in more complex settings like CVRPBLTW, masking must be applied with care.
>
>
> Following your suggestion, we have revised the wording in the main text and added detailed analysis and supporting figures in Appendix E.1 (Page 24-25, highlighted in pink).
>
> ```
> References:
> [1] Safety Representations for Safer Policy Learning. ICLR, 2025.
> [2] Safe Reinforcement Learning with Dead-Ends Avoidance and Recovery. IEEE Robotics and Automation Letters, 2023.
> ```

---

> ### Author Response · Authors · 2025-11-24
> **Response to Reviewer iA9w (4/4)**
>
> **\[R6: NeuOpt with short rollouts (Q2)\]** Thank you for the constructive suggestion. In our original paper, we provided the NeuOpt\* retrained with a short rollout budget ($T\_{train}$=5, same as CaR) in the third row of Table 5\. Following your suggestion, we have *added more experiments* on retraining NeuOpt\* by gradually decreasing the training horizon of NeuOpt and evaluating its performance under two inference budgets:  (1) $T\_{test}$=20 (same as CaR), and (2) $T\_{test}$=1,000, which reflects the refinement length typically required by improvement-based solvers.
>
> The results in Table R4 below show that **NeuOpt\* cannot produce any feasible solutions under a limited $T\_{test}$, regardless of whether its training horizon is short or long, while CaR achieves strong performance, with an optimality gap of 0.005% and 0% infeasibility within just 20 steps of refinement.** This discrepancy arises because NeuOpt\* is inherently unsuitable for short-horizon refinement. Even when retrained with our designed constraint-related features and constraint-aware penalty function in the MDP formulation, NeuOpt\* relies on long rollouts to gradually correct constraint violations and improve performance. In contrast, CaR is specifically designed to enable rapid feasibility refinement and solution improvement within tight step budgets. This highlights that short-horizon construct-and-refine is nontrivial and architecturally distinct from standard improvement-only methods. We have added this new experiment and discussion to the revised Appendix E.5, marked in pink for your ease of reference.
>
> **Table R4: Performance of NeuOpt\* with different short-horizon settings during training.**
>
> |Method|$T\_{train}$ | $T\_{test}$ | Gap | Infsb% |
> |-|-|-|-|-|
> |NeuOpt*|5 (same as CaR) | 20 | / | 100% |
> |NeuOpt*|10| 20 | / |100%|
> |NeuOpt*|20 | 20 | / |100%|
> |NeuOpt*|100| 20 | / | 100%|
> |NeuOpt*|200 (original NeuOpt) | 20| / | 100% |
> |NeuOpt*|5 (same as CaR) | 1000| / | 100% |
> |NeuOpt*|10|1000| 1.314% |99.79%|
> |NeuOpt*|20|1000|0.397% |7.36%|
> |NeuOpt*|100|1000|0.145% |0.22%|
> |NeuOpt*|200  (original NeuOpt) |1000| 0.061%|0.19%|
> |**CaR** |**5** | **20**|**0.005%**| **0.00%**|
> | | | | | |
>
> ---
>
> **\[R7: Sensitivity of different $p$ (Q4)\]** Thank you for the insightful question. As shown in the original Appendix E.8 (revised Appendix E.7), reducing $p$ from 10 to 5 on TSPTW-50 increases both infeasibility and optimality gap. To provide a more comprehensive analysis, we further retrain CaR with $p=20$ (see Table R5) and observe the expected trend: **larger $p$ yields better final performance, but the gains become increasingly marginal as $p$ grows.** Since $p$ is constrained only by GPU memory (we use a single Nvidia RTX 4090), we select the largest feasible $p$ in practice (i.e., $p=10$).
>
> **Table R5: Effects of different numbers of top-$p$ constructed solutions fed to the refinement module during training on TSPTW-50.**
>
> | $p$ | GPU memory | Gap | Infeasible% |
> | :---- | :---- | :---- | :---- |
> | 5 | 13.1GB | 0.032% | 0.33% |
> | 10 | 22.4GB | 0.014% | 0.01% |
> | 20 | 42.3GB | 0.007% | 0.00% |
> |  |  |  |  |
>
> Importantly, inference performance does not depend on $p$: during testing, each instance produces one solution, i.e, $S$=$p$=1, which is then refined. Therefore, CaR is not sensitive to $p$ at test time.
>
> We have included these new results and clarifications in the revised Appendix E.7 (marked in pink). We sincerely appreciate the reviewer’s thoughtful evaluation and constructive feedback. If there are still aspects of our explanation that appear ambiguous or incomplete, we would be glad to clarify them in more detail. We welcome any further questions or suggestions and are fully committed to improving the manuscript wherever needed.

---

### Official Review · Reviewer_wzGX · 2025-11-01

**Soundness:** 3
**Presentation:** 3
**Contribution:** 2
**Rating:** 6
**Confidence:** 3

**Summary:**

This paper presents Construct-and-Refine (CaR), a neural framework for efficiently handling hard constraints in vehicle routing problems (VRPs). By integrating construction and refinement paradigms via joint training and shared representations, CaR achieves significant improvements in feasibility, solution quality, and computational efficiency. The method demonstrates strong performance across diverse constrained VRP variants, outperforming both classical and neural baselines while maintaining minimal inference overhead.

**Strengths:**

1.	The paper is highly mature, with extensive experiments covering multiple VRP variants (TSPTW, CVRPBLTW, CVRP, TSPDL), scales, and constraint settings. The implementation details are thorough, including hyperparameters, baseline adaptations, and reproducibility measures.

2.	The paper provides detailed ablations validating key components, e.g., joint training, diversity loss, shared representations, and their individual impacts on performance. These analyses offer valuable insights into understanding the framework.

**Weaknesses:**

The technical contribution of this paper is limited. Although it addresses important constrained problems, CaR primarily adapts the existing construct-and-refine framework (e.g., NCS) to constrained domains. The shared representation and lightweight refinement design are incremental and do not introduce fundamentally novel ideas.

**Questions:**

1.	How does CaR achieve effective refinement in just 10 steps, while prior learn-to-improve methods require thousands? Is this solely due to the short-horizon design, or does the joint training framework play a critical role?

2.	Why are TSPDL results omitted from the main experiments? Additionally, Appendix results for TSPDL100 are missing. Could the authors clarify this gap?

---

> ### Author Response · Authors · 2025-11-24
> **Response to Reviewer wzGX (1/2)**
>
> We sincerely thank the reviewer for your time and effort in reviewing our paper. We greatly appreciate the recognition of our work as being highly mature, with extensive experiments, thorough implementation details, and informative ablation studies. We do hope the following response could address your remaining concern.
>
> ---
> **[R1: Clarification of Contribution (W1)]** We thank the reviewer for recognizing the importance of addressing constrained VRPs. While we understand that CaR and NCS both combine construction and improvement, we wish to clarify that *construct-and-refine* in CaR involves a fundamental redesign rather than adaptation compared to *construct-and-improve* in NCS.
>
> **Fundamentally, CaR and NCS employ opposing assumptions for different goals.** NCS aims to reduce optimality gaps under the assumption that feasibility is easily handled (via masking) and that very long runtimes are acceptable (e.g., tens of hours or days for 100-node CVRP/PDP). In contrast, CaR targets hard-constrained VRPs (TSPTW and CVRPBLTW) that current NCO solvers fail, and prioritizes efficiently, which is designed to operate under a limited runtime budget (less than a few minutes). The key differences are summarized as follows:
>
> **NCS: Infinite-horizon joint training for optimality**
>
> * **Goal & MDP:** NCS targets optimality improvement under the assumption of prolonged runtime. It formulates the problem as an infinite-horizon MDP trained with PPO.
> * **Shared component (Critic value):** Construction in NCS serves only to initialize solutions for long-term refinement. As both modules aim to reduce the optimality gap, NCS uses a shared critic for value estimation, but does not share policy network architectures (encoders/decoders) between modules and relies solely on simple feasibility masking, which fails under hard constraints.
>
> **CaR: short-horizon joint training for optimality and feasibility**
>
> * **Goal & MDP:** CaR targets efficiency-preserving constraint handling by formulating a short-horizon MDP that follows a “construct-then-fix” strategy in a few steps. To match this design, we remove the critic and apply REINFORCE, treating each refinement step equally. This allows us to optimize both optimality and feasibility by incorporating a penalty loss for both phases.
> * **Shared component (representation):** Unlike NCS, CaR uses a unified shared encoder, as hard constraints complicate embedding learning. This enables information synergy: constraint-aware representations learned during construction are reused in refinement. It creates a deep "feasibility awareness" unavailable in NCS, allowing the improvement module to understand the structural reasons for a construction failure and repair it efficiently.
>
> **Furthermore, these designs are task-specific.** Introducing NCS’s shared critic into CaR’s short-horizon MDP may introduce computational waste to learning another critic network without benefit. Conversely, applying CaR’s shared representation to NCS would likely fail due to imbalance: with NCS’s long improvement horizon, gradients would be dominated by the improvement module, washing out construction features. Critically, directly applying NCS to hard-constraint problems results in **100% infeasibility**.
>
> **CaR's Novelty:** We summarize below several ideas that, to the best of our knowledge, are **first attempts in the NCO literature** and have not been previously explored:
>
> - **A new constraint handling scheme**: beyond *feasibility masking* and implicit *feasibility awareness*, CaR proposes *explicit feasibility refinement*, which is a novel perspective that seeks learning-based repairing of infeasible solutions in a few post-construction steps while preserving optimality (see Table 1 for a conceptual comparison). This overcomes the limitations of existing methods, which often fail when complex constraints make one-shot valid construction exponentially difficult.
> - **An efficiency-preserving joint-training framework:** unlike LCP/NCS, CaR proposes an efficiency-preserving joint training framework that constructs diverse, high-quality solutions well-suited for rapid refinement, guided by our redesigned MDP and the tailored loss functions.
> - **A novel feasibility-awareness mechanism via shared representation**: We introduce a cross-module learning mechanism via shared encoders that strengthens constraint handling beyond standard feature or penalty-based methods. By enabling joint gradient flow, the model learns simultaneously from both "creation" and "fix" perspectives, generating deep, constraint-aware representations essential for complex problems.
> - **SoTA performance:** On TSPTW-100, CaR reduces the gap from 1.026% to 0.146% and infeasibility from PIP’s 4.47% to 0.02%, even outperforming LKH-3 (0.13% infeasibility). On CVRPBLTW-100, CaR achieves a -2.448% gap vs. OR-Tools, outperforming NeuOpt (39-51% infeasibility) and SGBS (0.854% gap). See Figure 3 for detailed comparisons.

---

> ### Author Response · Authors · 2025-11-24
> **Response to Reviewer wzGX (2/2)**
>
> **\[R2: How does CaR achieve effective refinement in just 10 steps? (Q1)\]**
> We appreciate the reviewer’s insightful question. We would like to clarify that CaR’s effectiveness under short-horizon refinement (\~10 steps) derives from more than just the short‑horizon MDP formulation; in contrast, it stems from the combination of strong initialization via the construction module and shared representations within our joint training framework.
>
> 1. In the original submission, we reported results for NeuOpt\* retrained under a short‑horizon MDP ($T\_{train}$ = 5, matching CaR) in Table 5\. To further address your question, we *add more experiments* on retraining NeuOpt\* by gradually decreasing the training horizon of NeuOpt and evaluating its performance under two inference budgets:  (1) $T\_{test}$=20 (same as CaR), and (2) $T\_{test}$​=1,000, which reflects the refinement length typically required by improvement-based solvers. Results in Table R1 show that the **no matter the training horizon, NeuOpt\* fails to produce any feasible solutions under the limited horizon $T\_{test}$=20,** while CaR achieves strong performance with a 0.005% optimality gap and 0% infeasibility within only 20 refinement steps.
>     **Table R1: Performance of NeuOpt\* with different short-horizon settings during training.**
>     |Method|$T\_{train}$ |$T\_{test}$ | Gap |Infsb%|
>     |-|-|-|-|-|
>     |NeuOpt*|5 (same as CaR)| 20 | / |100%|
>     |NeuOpt*|10| 20 | / |100%|
>     |NeuOpt*|20 | 20 | / |100%|
>     |NeuOpt*|100| 20 | / | 100%|
>     |NeuOpt*|200 (original NeuOpt) | 20| / | 100% |
>     |NeuOpt*|5 (same as CaR) | 1000| / | 100% |
>     |NeuOpt*|10|1000| 1.314% |99.79%|
>     |NeuOpt*|20|1000|0.397% |7.36%|
>     |NeuOpt*|100|1000|0.145% |0.22%|
>     |NeuOpt*|200  (original NeuOpt) |1000| 0.061%|0.19%|
>     |**CaR** |**5** | **20**|**0.005%**| **0.00%**|
>     | | | | | |
>
> 2. To verify that the joint training effect of CaR is non‑trivial, we compared against a naive baseline combining a pretrained PIP with NeuOpt (trained long or short). As shown in Table 5, the naive combination improved only marginally over PIP alone (0.172% vs. 0.177% gap, 2.59% vs. 2.67% infeasibility), suggesting that while high-quality initialization reduces the refinement burden, it is not sufficient.
> 3. In contrast, CaR’s joint training framework, featuring shared representations (see Table 6), a diversified construction policy (see Table 4), and other tailored components, enables effective multi-paradigm knowledge transfer, resulting in significantly better performance. The shared encoder allows the refinement policy to directly leverage constraint- and structure-aware representations learned during construction, which is critical for success under short-horizon constraints. As shown in Table 2, CaR achieves an optimality gap of just 0.005% and 0% infeasibility, clearly demonstrating that **CaR’s joint training framework is important for effective refinement within very limited rollout budgets.**
>
> We have added this new experiment and discussion to the revised Appendix E.5, marked in pink for your ease of reference. We sincerely appreciate the reviewer’s thoughtful evaluation and constructive feedback. If there are still aspects of our explanation that appear ambiguous or incomplete, we would be glad to clarify them in more detail. We welcome any further questions or suggestions and are fully committed to improving the manuscript wherever needed.
>
> ---
>
> **\[R3: Results of TSPDL-100 (Q2)\]**
> Thank you for the question. TSPDL was omitted in the main Table 2 due to its problem similarity to TSPTW. In our experimental design, we categorized VRPs into three representative types:
>
> 1. simple VRPs, where feasibility masking is tractable and effective (e.g., CVRP),
> 2. complex VRPs, where masking is tractable but overly restrictive, thus ineffective (e.g., CVRPBLTW),
> 3. complex VRPs, where computing feasibility masks is NP-hard or intractable (e.g., TSPTW, TSPDL).
>
> To maintain clarity and conciseness in the main table, we selected one representative from each category, i.e., TSPTW and CVRPBLTW. TSPDL was thus omitted purely for brevity.
>
> Regarding the TSPDL results, our initial submission reported TSPDL-50 in Table 3 to demonstrate that our method also applies to this setting. Following your suggestion, we have now added results for TSPDL-100 as well. However, *due to limited time and computing resources during the rebuttal period*, we trained CaR for only 2,000 epochs (compared with 10,000 epochs for PIP). Even without full training, the results below show that CaR consistently outperforms the state-of-the-art solver PIP. We will include the complete TSPDL-100 results in the updated Appendix once training is finished.
>
> **Table R2: Results of TSPDL.**
> |Method|TSPDL-50| |TSPDL-100| |
> |-|-|-|-|-|
> |  |Gap|Infsb%|Gap|Infsb%|
> |PIP|3.122%| 2.12%| 12.32% |7.91%|
> |**CaR-PIP**| **2.190%**| **0.26%**|**7.244%** (not fully trained)| **3.09%** (not fully trained)|
> | | | | | |

---

> > ### Comment · Reviewer_wzGX · 2025-11-27
> >
> > Thank you for your response. The additional experiments and analysis have addressed most of my concerns. I will maintain my positive score with higher confidence.

---

> > > ### Author Response · Authors · 2025-11-28
> > > **Thank you for your positive feedback and continued support!**
> > >
> > > We are delighted to have addressed your concerns. We truly appreciate your support and the time you dedicated to reviewing our work.

---

### Meta-Review · Area_Chair_RWWz · 2026-01-08

**Summary:**

The paper proposed Construct-and-Refine (CaR), a general framework for neural routing solvers designed to handle complex, hard constraints (e.g., TSPTW, CVRPBLTW) efficiently. CaR introduces an explicit feasibility refinement scheme via a joint training framework that couples a construction module with a lightweight refinement module, utilizing a unique shared representation to enhance feasibility awareness.

Reviewers and authors were actively involved in the rebuttal. The reviewers made substantial efforts to improve the draft, including clarifying conceptual novelty, expanding applicability, rigorous ablations, and deepening analysis on failure patterns. Overall, the paper is worthy to publish. For the camera-ready version, I encourage the authors to thoroughly go through the reviews and ensure every promised revision and new result is included.

**Reviewer Concerns:**

The AC has gone through the reviews and authors' responses. All concerns have been addressed.

**Reviewer Scores:**

[wzGX] & [FEWD]

Both reviewers explicitly confirmed they will maintain their positive scores of 6.

[8D1i]

This reviewer expressed satisfaction with the authors' detailed clarifications regarding theoretical analysis and applicability, confirming they will maintain the strong score of 8.

[iA9w]

While this reviewer did not participate in the discussion, the authors provided comprehensive responses addressing all critical concerns. Consequently, the score is likely to be maintained at 4 or potentially raised to 6.

---

### Decision · Program_Chairs · 2026-01-26

Accept (Poster)